# Concept Flow Models: Anchoring Concept-Based Reasoning with Hierarchical Bottlenecks

**Ya Wang**                                                                *ya.wang@fokus.fraunhofer.de*
*Fraunhofer Institute for Open Communication Systems*
*Freie Universität Berlin*

**Adrian Paschke**                                                        *adrian.paschke@fokus.fraunhofer.de*
*Fraunhofer Institute for Open Communication Systems*
*Freie Universität Berlin*

**Reviewed on OpenReview:** *https://openreview.net/forum?id=TNYLf65I3I&noteId=TNYLf65I3I*

## Abstract

Concept Bottleneck Models (CBMs) enhance interpretability by projecting learned features into a human-understandable concept space. Recent approaches leverage vision-language models to generate concept embeddings, reducing the need for manual concept annotations. However, these models suffer from a critical limitation: as the number of concepts approaches the embedding dimension, information leakage increases, enabling the model to exploit spurious or semantically irrelevant correlations and undermining interpretability. In this work, we propose Concept Flow Models (CFMs), which replace the flat bottleneck with a hierarchical, concept-driven decision tree. Each internal node in the hierarchy focuses on a localized subset of discriminative concepts, progressively narrowing the prediction scope. Our framework constructs decision hierarchies from visual embeddings, distributes semantic concepts at each hierarchy level, and trains differentiable concept weights through probabilistic tree traversal. Extensive experiments on diverse benchmarks demonstrate that CFMs match the predictive performance of flat CBMs, while substantially mitigating information leakage by reducing effective concept usage. Furthermore, CFMs yield stepwise decision flows that enable transparent and auditable model reasoning with hierarchical class structures.

## 1 Introduction

Deep Neural Networks have revolutionized many fields, yet their opaque decision-making poses significant challenges to reliability, interpretability, and traceability, especially in safety-critical applications (Adadi & Berrada, 2018; Rudin, 2019). Concept Bottleneck Models (CBMs) (Koh et al., 2020) offer a step toward interpretability by first predicting a set of human-understandable concepts, which are then used to predict the final label. The original CBMs (see Fig. 1) typically relied on feature extractors and manual concept annotations to supervise concept prediction during training. Despite improving interpretability, CBMs face two primary issues that limit their broader applicability: first, their reliance on manually labeled concepts restricts scalability; second, they are prone to information leakage (Margeloiu et al., 2021; Mahinpei et al., 2021; Havasi et al., 2022; Parisini et al., 2025), a phenomenon where learned concept representations encode unintended information about the task label or other concepts beyond their predefined semantics. This enables the model to achieve high accuracy by exploiting spurious correlations in concept activations rather than their semantic content, thereby compromising interpretability.

To address these limitations, post-hoc CBMs (Yuksekgonul et al., 2022) leverage pre-trained vision–language models (VLMs) to embed a set of task-specific linguistic concept descriptions. By comparing these concept embeddings with image feature embeddings, the model obtains concept activations, which are subsequently

weighted via a linear layer to generate the final prediction. In the remainder of this paper, we refer to this VLM-based variant as "CBM" for brevity, unless otherwise specified. While this approach removes the need for explicit concept learning and costly manual annotation, the problem of information leakage persists: pre-trained VLM embeddings inherently encode rich information beyond their intended concept semantics, and the linear layer may exploit or misuse these embeddings to achieve high accuracy. For example, when classifying images of cats, the model may rely on vehicle-related concepts such as "body paint" or "doors", which are unrelated to animals but spuriously correlate with the target class. Previous studies (Yan et al., 2023; Shang et al., 2024) have shown that as the number of concept embeddings approaches the feature embedding dimension, even random concepts—those entirely unrelated to the task—can allow the model to maintain high accuracy, thereby severely undermining interpretability.

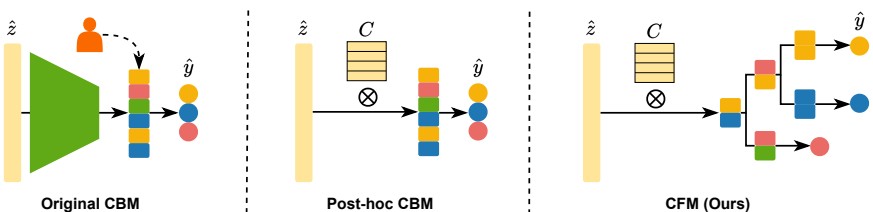

Figure 1: **Illustration of concept-based model structures:** Original Concept Bottleneck Models (left), Post-hoc Concept Bottleneck Models (center), and our proposed Concept Flow Models (right). Rounded rectangles represent concepts, trapezoids represent the feature extractor, circles denote classes, $\hat{z}$ indicates the feature embedding, and $C$ represents the concept matrix.

Recent works have sought to mitigate information leakage by discovering and selecting more compact concept sets (Yan et al., 2023; Yang et al., 2023; Schrodi et al., 2024; Shang et al., 2024) or by enforcing sparsity on the concept-to-label linear layer (Yuksekgonul et al., 2022; Oikarinen et al., 2023). These approaches aim to reduce the number of concepts onto which feature embeddings are projected during prediction, as fewer active concepts limit the model's capacity to exploit semantically irrelevant concepts, thereby reducing the risk of information leakage. To quantify this, Srivastava et al. (2024) introduced the *Number of Effective Concepts* (NEC), which measures the average number of concepts used to predict each class. However, as task complexity grows, the required number of concepts inevitably increases, heightening the risk of leakage. This raises the question: instead of projecting feature embeddings onto the entire concept space, can we partition the concept space into subspaces and route predictions only through the relevant ones, thereby isolating unrelated concepts from the decision path? Building on this intuition, we propose *Concept Flow Models* (CFMs) (see Fig. 1), which replace the flat bottleneck with hierarchical bottlenecks where each class prediction follows a unique path from root to leaf. This design decomposes prediction into a sequence of localized, concept-driven subproblems, offering three advantages: *reduced concept dimensionality at each node*—each step projects features onto a small, node-specific subset of concepts, limiting exposure to unrelated information; *reduced total concept usage per class*—each prediction aggregates concepts only along its decision path; and *traceable reasoning*—the tree structure yields step-by-step decision pathways, enhancing interpretability and enabling systematic auditing.

Our contributions are summarized as follows:

- We propose the Concept Flow Model (CFM), a concept-based model with hierarchical bottlenecks that decomposes decisions into step-by-step, interpretable paths. To support this, we develop a pipeline that constructs decision hierarchies using pretrained vision-language models, selects discriminative concepts without requiring manual concept annotation, and enables end-to-end model training.

- We provide both theoretical and empirical analysis demonstrating that CFMs mitigate accuracy gains from random, task-irrelevant concepts and increase reliance on meaningful semantic ones, while

matching the accuracy of flat CBMs under the same concept budget and structurally reducing the number of effective concepts per prediction, thereby reducing information leakage.

## 2 Related work

The paradigm of predicting intermediate concepts before final label prediction originated in early interpretability research (Kumar et al., 2009; Lampert et al., 2009), where attribute-based classifiers improved transparency at the cost of accuracy. Koh et al. (2020) formalized Concept Bottleneck Models (CBMs), demonstrating that concept-based prediction can enable human-in-the-loop interaction while maintaining competitive accuracy. Subsequent extensions improved CBM utility through concept editing, interactive mechanisms, and enhanced concept representations (Chauhan et al., 2023; Steinmann et al., 2023; Hu et al., 2024; Espinosa Zarlenga et al., 2022; 2023). Another direction models cross-concept dependencies within the bottleneck layer. Havasi et al. (2022) introduce autoregressive concept predictors that sequentially predict each concept conditioned on previous ones, capturing inter-concept correlations to improve concept accuracy and mitigate leakage. Vandenhirtz et al. (2024) propose stochastic formulations that propagate interventions through learned covariance structures, improving intervention effectiveness. Dominici et al. (2024) learn explicit causal graphs over concepts, enabling do-interventions and counterfactual reasoning for enhanced causal transparency. These approaches model dependencies within a flat concept layer, whereas our method achieves concept sparsity structurally by distributing concepts across a class hierarchy, a complementary mechanism that could potentially be combined with dependency modeling.

Despite these advances, they still rely on human-annotated concept sets, which limits scalability in domain-specific applications. To overcome this limitation, Yuksekgonul et al. (2022) leveraged multimodal models to generate concept embeddings from natural-language prompts, utilizing ConceptNet (Speer et al., 2017) and predefined concept sets to reduce manual annotation effort. Later works (Oikarinen et al., 2023; Yang et al., 2023) improved scalability by generating domain-specific concepts with large language models, enabling open-vocabulary concept acquisition. However, these approaches face a critical challenge (Yan et al., 2023; Shang et al., 2024; Srivastava et al., 2024): as concept count grows toward the embedding dimension, even random concepts can induce linear separability, allowing models to preserve accuracy through spurious correlations rather than semantic grounding. Recent efforts focus on selecting or discovering high-quality concepts using Elastic Net regularization (Yuksekgonul et al., 2022), submodular selection (Yang et al., 2023), incremental discovery (Shang et al., 2024), dictionary learning (Yan et al., 2023) and open-domain object detectors (Srivastava et al., 2024). Nonetheless, the required concept count inevitably grows with task complexity. Our method introduces a human-aligned decision hierarchy to decompose the task, which structurally reduces per-node concept demand and enforces hierarchical sparsity constraints, thereby mitigating information leakage and enhancing the utility of semantic concepts in classification.

Another line of work enhances interpretability through hierarchical structures. Some models dynamically route inputs through tree-like architectures (Murdock et al., 2016; Mullapudi et al., 2018; Cai et al., 2021), but rely on impure leaves and yield stochastic, uninterpretable paths. Others (Deng et al., 2014; Redmon & Farhadi, 2017; Brust & Denzler, 2019) use predefined hierarchies like WordNet (Miller et al., 1990), which introduce biases when conceptual similarity misaligns with visual discriminativeness. Neural-Backed Decision Trees (NBDTs) (Wan et al., 2020) train networks with path probabilities and extract hierarchies by clustering final-layer weights, assigning node labels using WordNet. While our method shares a similar training objective to NBDT, it differs by incorporating concept-level semantics directly into the decision nodes and learning the tree hierarchy end-to-end, enabling inherently interpretable, concept-driven inference paths. Recent work has explored combining decision trees with concept-based models for different purposes. Rodríguez et al. (2024) replace the CBM label predictor with a differentiable soft decision tree to enhance decision transparency, though each node receives the full concept vectors and all leaves contribute probabilistically to predictions. Ragkousis & Parbhoo (2024) use decision trees to inspect and quantify information leakage by comparing hard versus soft concept representations, enabling localized leakage control but requiring ground-truth concept annotations. Other approaches introduce hierarchies within concept representations rather than over classes: Sun et al. (2024) augment perceptual concepts with descriptive attributes using class-specific subsets and an intervention matrix, while Pittino et al. (2023) organize concepts into multiple categories whose features are concatenated for final classification, allowing all concept types to jointly influence

predictions. In contrast, our method constructs a hierarchy over classes, distributes concepts to internal nodes such that each prediction uses only path-specific subsets, and operates without manual concept annotation, providing structural isolation that complements these approaches.

## 3 Method

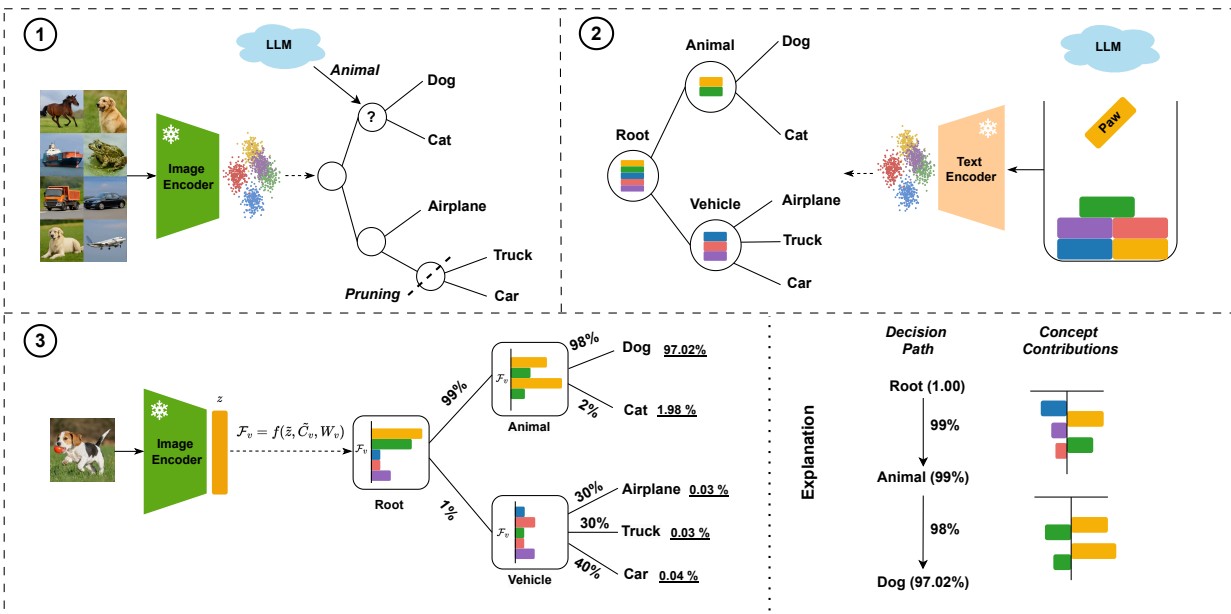

Figure 2: **Building Concept Flow Models:** Our framework consists of three phases: (1) *Hierarchy Extraction & Semantic Annotation*: Construct a decision hierarchy from CLIP embeddings, prune low-separation nodes, and label internal nodes using LLMs. (2) *Concept Generation & Allocation*: Generate LLM-based candidate concepts, select discriminative ones, and allocate them to hierarchical bottlenecks. (3) *Differentiable Concept Flow Training*: Train the model end-to-end with hierarchical path traversal loss, enabling traceable predictions through concept-weighted transitions.

As illustrated in Fig.2, our approach differs from Concept Bottleneck Models, which expose all concepts in a flat bottleneck layer. Instead, we extract a semantically coherent tree hierarchy from CLIP embeddings (Sec.3.2) and allocate LLM-generated concepts to its internal nodes (Sec.3.3), train them with concept-driven probabilistic flows (Sec.3.4), thereby constraining each class prediction to rely primarily on concepts along the most probable decision path, while reducing the influence of unrelated concepts and yielding traceable, interpretable predictions.

### 3.1 Problem setup

We consider a multi-class classification task over a dataset $\mathcal{D} = \{(x, y)\}$ where $x \in \mathcal{X}$ denotes input images and $y \in \mathcal{Y} = \{1, \ldots, |L|\}$ represents class labels. The core challenge lies in constructing an interpretable decision tree hierarchy $T = (V, E)$ where leaf nodes $L \subset V$ correspond to target classes $\mathcal{Y}$, while internal nodes encode hierarchical relationships by representing localized subspaces of the overall concept space. Unlike CBMs that compress decisions through a single layer with uncoordinated concepts, we distribute discriminative concepts across multiple hierarchical nodes. Each internal node $v \in V \setminus L$ maintains: (1) a set of human-interpretable concepts, (2) learned importance weights for these concepts, and (3) transition probabilities to child nodes based on concept relevance. The tree constrains each class to follow a unique decision path from the root to the corresponding leaf node, thus minimizing the influence of concepts assigned to nodes outside this path. The learning objective minimizes the cross-entropy loss through concept-weighted hierarchy traversal:

$$\min_{\theta} \mathbb{E}_{(x,y)\sim\mathcal{D}} \left[ -\log p_{\theta}(y|x) \right], \tag{1}$$

where $\theta$ denotes all concept importance weights in the hierarchy, and $p_{\theta}(y|x)$ represents the probability of reaching the ground-truth class $y$ through concept-driven transitions (detailed in Sec. 3.4).

## 3.2 Tree hierarchy extraction

Our approach builds on a pretrained VLM, CLIP (Radford et al., 2021), which comprises an image encoder $\Phi_I : \mathcal{X} \to \mathbb{R}^d$ and a text encoder $\Phi_T : \mathcal{T} \to \mathbb{R}^d$, projecting inputs from both modalities into a shared $d$-dimensional semantic space. The cross-modal alignment of CLIP ensures that semantically related images and texts are embedded closely together, resulting in a structured representation space that supports both hierarchical clustering and concept-based reasoning (Bhalla et al., 2024; Stein et al., 2024). We leverage this property as a unified basis for constructing the concept hierarchy and performing concept selection. To construct a semantic hierarchy, we first compute class-level embeddings $\boldsymbol{\mu}_i$ by averaging the CLIP image embeddings of all instances belonging to class $i$. We then apply agglomerative clustering with Ward's linkage (Ward Jr, 1963) to the resulting centroids $\mathcal{M} = \{\boldsymbol{\mu}_1, \ldots, \boldsymbol{\mu}_{|L|}\}$, yielding a binary tree represented by a linkage matrix $Z \in \mathbb{R}^{(|L|-1)\times 4}$. The hierarchy $T$ is built in a bottom-up manner by iteratively merging node pairs $(v_p, v_q)$ with the smallest merge distance in $Z$. Overly deep trees can cause an exponential growth in node count, fragmenting the concept space into small subspaces and yielding weakly separated nodes near the leaves. We mitigate this with distance-based pruning, removing nodes whose merge distances fall below the $t$-th percentile of all distances in $Z$, producing a multi-branch hierarchy with fewer, broader internal nodes. Alternatively, node separability can be measured by the classification accuracy of a Linear Discriminant Analysis (LDA) fitted on its samples, pruning nodes whose scores fall below a threshold.

**Semantic annotation:** We annotate internal nodes via breadth-first traversal using LLM prompts. For each non-leaf node, we generate descriptive labels by prompting an LLM with the leaf classes in its subtree, producing hierarchical summaries. Since tree structures vary with VLM backbones and pruning thresholds, manual review may be applied to ensure the selected backbone is well-suited to the target domain, as different VLMs are trained on different data and exhibit varying strengths across tasks. In practice, widely-used VLMs (e.g., CLIP ViT-B/32) produce semantically coherent hierarchies for common domains such as general object or action recognition; manual intervention is primarily needed for specialized domains where off-the-shelf VLMs may lack sufficient coverage (see Appendix A.1).

## 3.3 Concept selection and distribution

Following prior works (Yan et al., 2023; Yang et al., 2023), we leverage LLMs to generate candidate concepts for each class, enhancing visual and hierarchical grounding by including image samples and hierarchical context in the prompt. For each internal node $v$, we aggregate all candidate concepts from the leaf classes under that node to form its concept set. We preprocess this aggregated set by discarding concepts that contain class names and eliminating semantically redundant concepts (see Appendix A.2 and A.3). The resulting filtered concepts $\mathcal{E}_v = \{e_{v,1}, \ldots, e_{v,n_v}\}$ are embedded via the text encoder $\Phi_T$ into a matrix $C_v \in \mathbb{R}^{n_v \times d}$. To evaluate concept relevance for node $v$, we compute the cosine similarity between all training image features $Z = \Phi_I(\mathcal{X}) \in \mathbb{R}^{n \times d}$ and the concept embeddings $C_v$:

$$S_v = \tilde{Z}\tilde{C}_v^{\top} \in \mathbb{R}^{n \times n_v}, \tag{2}$$

where $\tilde{Z}$ and $\tilde{C}_v$ denote the $\ell_2$-normalized rows of $Z$ and $C_v$, respectively. For large-scale datasets, computing $S_v$ over all $n$ samples can be prohibitive. Since the goal is concept selection rather than model training, a representative subset suffices. We therefore apply stratified sampling by grouping samples by class and randomly drawing a balanced subset from each class to bound the total samples (e.g., $n_{\max} = 15,000$) while ensuring sufficient samples per class (e.g., 30) for reliable estimation. We then perform concept selection for each internal node $v$ using Lasso-regularized multi-class logistic regression. Let $W \in \mathbb{R}^{n_v \times |L|}$ denote the

weight matrix and $y_{il} = \mathbb{I}[y_i = l]$ the class indicator. We optimize:

$$\min_W \; \sum_{l=1}^{|L|} \sum_{i=1}^n \mathcal{L}(y_{il}, W_l^\top s_i) + \lambda \sum_{l=1}^{|L|} \|W_l\|_1, \tag{3}$$

where $s_i$ is the $i$-th row of $S_v$. Each concept $e_{v,j}$ receives an importance score $I_j = \sum_{l=1}^{|L|} |W_{lj}|$, and the top $r$ concepts with highest $I_j$ are selected to form the final set $C_v^r$. To determine $r$, we adopt a hybrid allocation strategy based on the node child count and the average concept share per node $B$ (see Appendix A.4):

$$r = \max\bigl(\alpha \, |\mathrm{Ch}(v)| + (1 - \alpha) \, |\mathrm{B}|, \; r_{\min}\bigr), \tag{4}$$

where $\alpha \in [0, 1]$ and $r_{\min}$ are adjusted based on tree balance, preventing any node from being underrepresented or overburdened with concepts. For simplicity, we omit the superscript $r$ in $C_v^r$ hereafter.

## 3.4 Model architecture and training

CFM replaces the flat bottleneck layer with a hierarchy of bottleneck nodes, each equipped with selected concepts and trainable weights. This enables a differentiable decision process that propagates class probabilities along the concept-driven hierarchy. Each internal node $v$ is parameterized by two matrices:

$$C_v \in \mathbb{R}^{r \times d}, \quad W_v \in \mathbb{R}^{r \times m}, \tag{5}$$

where $C_v$ represents concept embeddings pre-selected for node $v$, and $W_v$ denotes trainable weights for these concepts, which model the local branching decisions at each node.

**Temperature-scaled transitions:** Each node receives a normalized image embedding $\tilde{z} = \Phi_I(x)/\|\Phi_I(x)\|_2$ as input. It first projects the embedding onto the pre-selected concept matrix $\tilde{C}_v \in \mathbb{R}^{n_v \times d}$ (as defined in Section 3.3), obtaining concept activations. These activations are then transformed by the learned local concept weight matrix $W_v \in \mathbb{R}^{n_v \times m}$, where $m = |\mathrm{Ch}(v)|$ is the number of child nodes, yielding the transition probabilities to the child nodes. To control the sharpness of branching decisions, each node also learns a temperature parameter $T_v \in \mathbb{R}^+$. The transition probability from node $v$ to one of its children $v_j$ is computed as:

$$p_{v \to v_j}(x) = \mathrm{softmax}\left( \frac{(\tilde{z}\tilde{C}_v^\top) W_v}{T_v} \right)_j. \tag{6}$$

In contrast to CBMs, which connect input embeddings to all concepts globally, CFMs associate each node with a local subset of concepts, thereby enabling parallel computation across nodes with no significant increase in computational overhead (see detailed complexity analysis in Appendix C.1).

**Path probability aggregation:** The class probability corresponds to the probability of reaching leaf node $l \in L$ from the root node $v_0$, computed recursively along the path $\pi(v_0, l)$:

$$p_{v_0 \to l}(x) = \left( \prod_{(v \to v_j) \in \pi(v_0, l)} p_{v \to v_j}(x) \right) \cdot b_l, \tag{7}$$

where $b_l \in \mathbb{R}^+$ is a learnable bias term calibrating path-specific confidence. Final predictions use:

$$\hat{y}(x) = \arg\max_{l \in L} \; p_{v_0 \to l}(x). \tag{8}$$

**Training objective:** We freeze the CLIP image encoder $\Phi_I$ and optimize the tree parameters $\{W_v, T_v, b_l\}$ by minimizing the negative log-likelihood of ground-truth class probabilities:

$$\mathcal{L} = -\frac{1}{|\mathcal{D}|} \sum_{(x,y) \in \mathcal{D}} \log p_{v_0 \to y}(x), \tag{9}$$

where $p_{v_0 \to y}(x)$ is the probability of the hierarchical decision path from the root $v_0$ to the leaf node corresponding to the true class $y$.

# 4 Theoretical Analysis of Concept Usage in CFMs

We provide a theoretical analysis of CFMs, focusing on how they utilize concepts differently from flat CBMs. Our analysis examines two scenarios: when models use *random concepts*, which are unrelated to the prediction task (either sampled i.i.d. from the unit sphere in theory or drawn from a general English dictionary in experiments); and when both models use curated *semantic concepts*, which are semantically related to the target task (e.g., "has wings" for birds). Let $(x, y) \sim \mathcal{D}$ be the data distribution, and let $z = \Phi_I(x) \in \mathbb{R}^d$ denote the CLIP image embedding. We assume a total concept budget of $R$ vectors for both CBM and CFM.

**CBM.** A flat CBM uses a single concept matrix $C \in \mathbb{R}^{R \times d}$ to compute logits $s = \tilde{z}\tilde{C}^\top$, followed by classification $\hat{y} = \arg\max_l (sW)_l$.

**CFM.** A CFM constructs a tree with $m$ internal nodes. Each node $v$ is assigned a subset of $r'$ concepts from a total of $R$ available concepts, where the expected number of concepts per node satisfies $r' \approx R/m$. Each node also contains local branching weights $W_v$. During inference, each input traverses a root-to-leaf path $\pi(x) = (v_1, \ldots, v_\ell)$ of expected length $\ell = \mathbb{E}[\ell] \leq \log_b m$, where $b$ denotes the average branching factor of the tree.

## 4.1 Information-Leakage Barrier of Hierarchical Bottlenecks

**Proposition 4.1** (Leakage Resistance). *Let the image embeddings $\mathcal{Z} = \{(z_i, y_i)\}_{i=1}^n \subset \mathbb{R}^d \times [|L|]$ be linearly separable across $|L|$ classes. Using random concept vectors:*

  (i) **CBM.** *Separability is still preserved with high probability once the total concept count satisfies $R = \Omega(d)$.*

  (ii) **CFM.** *To keep* every *internal node separable, each node must receive $r' = \Omega(d)$ concepts, giving a total requirement $R = \Omega(md)$ to ensure global separability, i.e., a factor $m$ larger than the flat CBM.*

**Interpretation.** A flat bottleneck leaks all $d$ dimensions at once, so $\Omega(d)$ random projections suffice to reconstruct the decision boundary to preserve accuracy. In a hierarchical structure, however, each of the $m$ internal nodes handles a different local classification subproblem. Since random projections do not capture specific properties of the data, a small set of random concepts cannot reliably separate subproblems. Thus, each node requires its own set of $\Omega(d)$ random concepts, resulting in a total of $\Omega(md)$ concepts to ensure global separability. Moreover, as $m$ increases under a fixed concept budget $R$, each node receives fewer concepts, reducing concept sharing compared to a CBM, which shares all $R$ concepts in a single layer. Distributing concepts across more nodes naturally enforces concept isolation: each predicted class relies only on concepts along its decision path, increasing concept sparsity and mitigating information leakage).

## 4.2 Semantic Efficiency of Hierarchical Decision Paths

**Proposition 4.2** (Semantic Concept Sparsity via Hierarchical Structure). *Assume that for each internal node there exists a small concept subset of size $r'$ that linearly separates its child clusters with high probability. If the global budget $R$ is distributed so that each node receives at least this $r'$, then:*

  (i) *The resulting CFM can achieve comparable training accuracy to a flat CBM that uses all $R$ concepts at once.*

  (ii) *The number of concepts contributing to a single prediction is at most $\ell r'$, where $r' \approx R/m$ and $\ell$ is the root-to-leaf path length. For a balanced tree, $\ell \approx \log_b m$, so the expected concept usage per prediction is $O\left(\frac{R}{m} \log_b m\right)$, compared to $O(R)$ for the flat model.*

**Interpretation.** Assuming that the tree hierarchy reflects the natural semantic hierarchy of the data, each internal node only needs to separate its own (smaller) set of child clusters, which simplifies the local separation task and allows each node to use only a modest subset of concepts ($r' \ll R$) instead of all $R$ concepts. For a

single prediction, only the concepts along the root-to-leaf decision path are used. Thus, the expected number of concepts per prediction is at most $\ell r'$. In practice, this is much less than $R$, since irrelevant concepts (outside the decision path) are structurally isolated. Mathematically, the upper bound $O\left(\frac{R}{m}\log_b m\right)$ shows that the ratio $\frac{\log_b m}{m}$ is less than 1 for any $m > 1$ and $b > 1$, and satisfies $\lim_{m \to \infty} \frac{\log_b m}{m} = 0$. Thus, as the hierarchy becomes more granular (i.e., as $m$ increases), the number of used concepts per prediction decreases rapidly, promoting greater concept sparsity and reducing the risk of information leakage.

**Why accuracy need not drop.** When the hierarchy is semantically coherent—e.g., constructed via hierarchical clustering of CLIP embeddings, with nodes of low separability pruned—each internal node handles a simpler "few-way" sub-problem. The target class only consults concepts within its decision path; other concepts structurally do not activate for the prediction. A handful of curated semantic concepts can be shared by all leaf classes under the same node, supporting accurate decisions. In contrast, random concepts lack semantic alignment and cannot be effectively reused by its associated classes.

**Semantic Utility Metric.** To complement the above analysis, we introduce the *Semantic Improvement over Random Concepts (SIR)* metric to quantify the role of semantic concepts in mitigating information leakage. SIR measures the relative accuracy gain when replacing random concepts with curated semantic ones under the same concept budget:

$$\Delta = \frac{\text{acc}_{\text{sem}} - \text{acc}_{\text{rand}}}{\text{acc}_{\text{rand}}} \quad (\times 100\%), \tag{10}$$

where $\text{acc}_{\text{sem}}$ and $\text{acc}_{\text{rand}}$ denote model accuracy with semantic and random concept sets, respectively. A high $\Delta$ indicates strong reliance on task-relevant features, while a low or zero value suggests that predictions can be replicated with uninformative concepts—indicating potential leakage.

## 5 Experiments

To empirically validate our theoretical claims in Propositions 4.1 and 4.2, we conduct experiments examining the two scenarios analyzed in our theory: using non-semantic random concepts and curated semantic concepts. We further perform ablation studies to assess the contribution of individual components within CFMs, visualize and interpret the decision-making process, and discuss broader implications and limitations.

### 5.1 Setup

**Datasets.** We conduct our evaluations on five image classification benchmarks: CIFAR-10, CIFAR-100 (Krizhevsky et al., 2009), UCF-101 (Soomro et al., 2012), CUB-200 (Wah et al., 2011), and TinyImageNet (Le & Yang, 2015). These datasets span a range of semantic granularity and domain complexity. Specifically, CIFAR-10 contains 10 general object classes, CIFAR-100 and UCF-101 include 100 classes each (with CIFAR-100 focused on common objects and UCF-101 on action scenes), and both CUB-200 and TinyImageNet contain 200 fine-grained categories, with the former targeting bird species classification and the latter composed of small-scale versions of ImageNet classes. Additional experiments on ImageNet-1K are provided in Appendix B.8.

**General settings.** Unless otherwise specified, all models use the CLIP ViT-B/32 backbone outputting 512-dimensional embeddings. We train with Adam (lr=0.01, batch size=128) using ReduceLROnPlateau (learning rate halved on validation loss plateaus) and early stopping with a 20% validation split. Training terminates upon validation stagnation, retaining the best checkpoint. For controlled experiments, CFMs are pruned to four internal nodes with 3–4 layers in depth (see Appendix B.2 for details).

### 5.2 Scenario 1: How do CFM and CBM perform under random, non-semantic concepts?

**Settings.** We systematically compare CFM and CBM on CIFAR-10 using random (non-semantic) concepts, and include a linear probe model (Radford et al., 2021) as an upper bound of the training accuracy (trained directly on CLIP image embeddings without interpretability constraints). Random concepts are generated

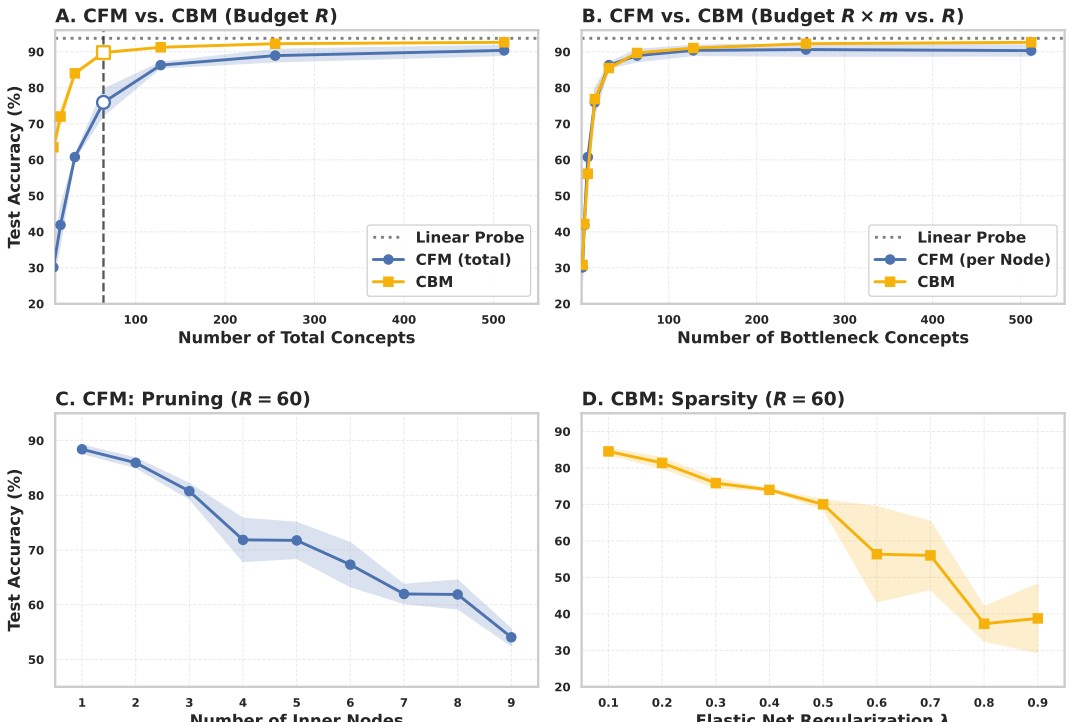

Figure 3: **Information Leakage with Random Concepts.** (**A**) Accuracy of CFM and CBM as total concept budget $R$ increases (same $R$ for both). (**B**) Accuracy when CFM receives $R$ concepts per node ($R \times m$ total), compared to CBM with $R$. (**C**) Effect of the number of internal nodes on CFM accuracy at $R = 60$. (**D**) Impact of Elastic Net regularization on CBM accuracy at $R = 60$. All experiments are repeated three times with different random seeds. Detailed numerical results are provided in Appendix B.4.

by sampling dictionary phrases (see Appendix A.2). Our experiments examine: (A) accuracy as the total concept budget $R$ increases, with both models constrained to the same $R$; (B) accuracy when CFM receives $R$ concepts per node (i.e., $R \times 4$ concepts in total), compared to CBM with $R$; (C) varying the number of internal nodes in CFM under a fixed $R = 60$; and (D) varying the sparsity regularization strength $\lambda$ in CBM (implemented via Elastic Net from Zou & Hastie (2005) with L1 ratio $\alpha = 0.9$) under a fixed $R = 60$.

**Results.** In (A), CBM accuracy rises quickly with $R$, reaching 90% with only 64 concepts and approaching the linear probe baseline as $R$ approaches the embedding dimension, consistent with Proposition 4.1(i). In contrast, CFM achieves only 75% under the same budget, reflecting its structural resistance to information leakage from random concepts. In (B), when CFM's total budget is increased to $R \times 4$, its accuracy curve closely tracks that of CBM, consistent with Proposition 4.1(ii), which states that achieving global separability in CFM requires scaling the total concept count with the number of internal nodes. In (C), reducing the number of internal nodes by pruning relaxes the structural sparsity constraints, increasing accuracy as more concepts are shared per node and information isolation is weakened. The complete unpruned binary tree represents the case of most strict concept isolation. In (D), increasing sparsity regularization in CBM generally reduces accuracy by limiting the number of effective concepts, analogous to the structural sparsity imposed by CFM, although sparsity regularization cannot structurally prevent misuse of concepts from other classes. Notably, sparsity regularization can similarly be applied to CFM to control concept usage (see Appendix B.3). Overall, these results confirm that CBMs with a flat structure are prone to information leakage, while CFMs provide a structural barrier that limits accuracy gains from random concepts, consistent with our theoretical analysis.

### 5.3 Scenario 2: How do CFM and CBM perform under curated semantic concepts?

**Settings.** We compare CFM and CBM using curated semantic concepts, with the linear probe again serving as a non-interpretable baseline. For CBMs, we include two representative models: PCBM (Yuksekgonul et al., 2022), which retrieves predefined concepts from ConceptNet (since the original PCBM lacks concept selection, we implement it by ranking concepts based on ConceptNet edge weights and selecting the top-$r$ most relevant), and Labo (Yang et al., 2023), which uses LLMs to generate and select task-specific concepts. To quantify the utility of semantic concepts, we further compare each semantic model to its random counterpart: CBM (Random) and CFM (Random), where semantic concepts are replaced with random ones. For fair comparison, CFM uses the same candidate concept set as Labo, and all models are trained without sparsity regularization. We conduct experiments on all five datasets, keeping the total concept budget identical across models. Specifically, the total concept budget is set to two concepts per class for each dataset, except for CIFAR-10, which uses a fixed budget of 60 concepts. This yields 60, 200, 202, and 400 concepts for CIFAR-10, CIFAR-100, UCF-101, and CUB-200/TinyImageNet, respectively. For CFMs, we fine-tune the pruning threshold and concept allocation ratio to obtain exactly four internal nodes with suitable concept distribution. Hyperparameters for Labo and PCBM are also optimized on each dataset (see Appendix B.2). Additional experiments comparing CFM and PCBM with sparsity regularization are provided in Appendix B.6.

**Metrics.** In addition to accuracy, we report NEC (the Number of Effective Concepts) proposed by Srivastava et al. (2024), which quantifies the average number of concepts used to predict each class. We also report the Semantic Improvement over Random Concepts (SIR), which measures the accuracy gain achieved by semantic concepts relative to random concepts.

**Results.** Consistent with Proposition 4.2, CFMs achieve comparable accuracy to CBMs (PCBM and Labo) under the same concept budget, while using far fewer effective concepts (see Table 1). For example, on CIFAR-10, CFMs reach 91.59% accuracy with only 7 effective concepts, compared to Labo (90.93%) and PCBM (91.15%), both using around 57. On TinyImageNet, CFMs achieve 70.79% accuracy with 92 effective concepts, outperforming Labo (69.86%) and achieving comparable results to PCBM (71.07%), both using over 360. This efficiency stems from structural constraints that limit each prediction to the concepts along its root-to-leaf path, resulting in much lower average concept usage per prediction. As a result, CFMs show substantially higher SIR across all datasets, for instance, 19.78 on CIFAR-10 versus 1.94 for Labo, with similar gains on TinyImageNet and other benchmarks. This structural sparsity constraint also reduces information leakage, as predictions depend on a smaller, path-specific subset of concepts rather than the entire pool. Additionally, CFMs' much lower accuracy with random concepts (e.g., 76.46% on CIFAR-10 vs. CBM's 89.20%) highlights their reliance on meaningful semantic concepts. When the concept budget is limited, CFMs consistently match CBM accuracy by prioritizing semantic utility.

Table 1: Comparison of various methods across five benchmark datasets. We report the number of effective concepts (NEC), classification accuracy (Acc), and semantic improvement over random concepts (SIR). Results are reported as mean over 3 runs. Bold values highlight the best in each column (excluding linear probe). Full results with standard deviations are provided in Appendix B.5.

| Method | CIFAR-10 | | | CIFAR-100 | | | UCF101 | | | CUB200 | | | TinyImageNet | | |
|---|---|---|---|---|---|---|---|---|---|---|---|---|---|---|---|
| | NEC | Acc | SIR | NEC | Acc | SIR | NEC | Acc | SIR | NEC | Acc | SIR | NEC | Acc | SIR |
| Linear Probe | N/A | 94.59 | N/A | N/A | 77.53 | N/A | N/A | 95.00 | N/A | N/A | 71.68 | N/A | N/A | 75.25 | N/A |
| CBM (Random) | 56.80 | 89.20 | 0 | 184.20 | 68.50 | 0 | 187.31 | 79.15 | 0 | 344.19 | 52.23 | 0 | 362.34 | 68.78 | 0 |
| CFM (Random) | 6.90 | 76.46 | 0 | **54.24** | 62.00 | 0 | **44.99** | 72.76 | 0 | **104.61** | 47.08 | 0 | **89.83** | 64.89 | 0 |
| PCBM | 56.80 | 91.15 | 2.19 | 183.99 | 68.98 | 0.70 | N/A | N/A | N/A | N/A | N/A | N/A | 362.01 | **71.07** | 3.32 |
| Labo | 57.73 | 90.93 | 1.94 | 184.33 | 70.98 | 3.62 | 190.60 | **84.62** | 6.91 | 357.92 | 65.13 | 24.70 | 369.36 | 69.86 | 1.57 |
| CFM (Ours) | **6.90** | **91.59** | **19.78** | 55.02 | **72.29** | **16.61** | 45.80 | 84.60 | **16.27** | 109.34 | **65.68** | **39.50** | 92.12 | 70.79 | **9.09** |

### 5.4 Ablation Study

**Effect of tree hierarchy on performance.** To validate the effectiveness of our CLIP-induced hierarchy generation strategy, we compare it against two alternative approaches: (1) Random, which generate the hierarchy by randomly clustering the classes, and (2) NBDT (Weights), following Wan et al. (2020), which generate the hierarchy by clustering the final-layer weights (each row of the final-layer weight matrix

corresponds to a class representative) in the trained linear probe model (see Appendix A.1). As shown in Table 2, our approach achieves higher accuracy on four out of five datasets, with particularly strong gains on UCF-101 (+3.16% vs. Weights) and CIFAR-10 (+1.26%). By contrast, the random baseline consistently ranks last, highlighting the importance of semantic coherence in hierarchy construction. The CLIP-induced hierarchy underperforms on CUB-200, likely due to the fine-grained nature of bird-species classification, which challenges CLIP's general-purpose embeddings.

Figure 4 shows that, when the total concept budget is fixed, increasing tree depth reduces accuracy. This is because the number of internal nodes grows exponentially with depth, so each node receives fewer concepts, weakening local separability and accumulating errors along the decision path (see Table 4 for further evidence). Pruning the internal nodes mitigates this effect by allocating more concepts per node and improving accuracy. The number of internal nodes thus acts as a structural sparsity constraint: deeper trees promote interpretability through sparser, more traceable decision paths, but risk degrading accuracy if concepts or the CLIP backbone are not sufficiently strong. In practice, tree pruning serves as a sparsity hyperparameter that requires tuning.

Table 2: Accuracy (%) comparison among three hierarchy-construction strategies: Random, NBDT (Weights), and our CLIP-induced approach. Each method builds tree hierarchies differently before concept selection. Bold values denote the highest accuracy per dataset.

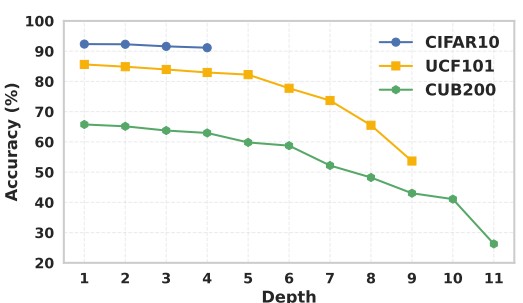

Figure 4: Impact of tree depth pruning

| Method | CIFAR-10 | CIFAR-100 | UCF101 | CUB200 | TinyImageNet |
|---|---|---|---|---|---|
| Random | 90.82 | 69.97 | 81.53 | 60.44 | 60.35 |
| NBDT | 91.00 | 71.13 | 82.28 | **66.24** | 70.84 |
| Ours | **92.26** | **72.16** | **85.44** | 65.79 | **71.24** |

**Evaluation of concept selection strategies.** We compare our Lasso regression-based concept selection method against four baselines (see more detail in Appendix A.5): random selection, CLIP similarity maximization, concept embedding orthogonality, and submodular optimization (Yang et al., 2023). As Table 3 shows, our method outperforms all alternatives, achieving 92.26% on CIFAR-10 (vs. 87.18% for submodular) and 65.79% on CUB-200 (vs. 61.63% for orthogonality). The poor performance of similarity-based selection (83.79% on CIFAR-10) suggests that naively prioritizing concept-image relevance leads to redundant concepts, while orthogonality over-constrains the concept space. Submodular optimization, which greedily selects discriminative and diverse concepts, outperforms both approaches. Nonetheless, our method better balances diversity and discriminative strength, yielding compact yet semantically complementary concept sets.

**Impact of model architecture design choices.** Table 4 ablates CFM's core components. Removing the concept matrix (*w/o Concept Matrix*), which approximates the NBDT design, nearly matches Linear Probe's accuracy (94.67% vs. 94.75% on CIFAR-10), indicating the hierarchy itself contributes minimally to accuracy loss - the primary limitation stems from insufficient concepts across nodes in deeper hierarchies. Path calibration contributes measurably to performance (81.72% vs. 85.44% on UCF101 without it). Node temperature scaling contributes most to fine-grained tasks (62.62% → 65.79% on CUB-200), moderating error propagation in deep hierarchies. Adding a hierarchical auxiliary loss slightly degrades performance (91.13% vs. 92.26%), consistent with the observations by Wan et al. (2020) that over-supervision at each node disrupts path probability learning (see Appendix A.6).

## 5.5 Visualization and interpretation of concept-driven decision paths

We provide qualitative examples comparing the decision paths and activated concepts of CFMs and CBMs. The top part of Fig. 5 shows an example of a "cat" from CIFAR-10. The CLIP-induced hierarchy in CFM first separates categories into "Animal" and "Vehicle", and then further narrows the scope to "Domesticated" animals. At each stage, CFM uses the current node's selected concepts to determine the transition probabilities to child nodes (i.e., the next concept subspaces), effectively routing the prediction and structurally isolating concepts outside decision paths. For the cat sample, CFM uses "facial markings" and "fur texture" to identify

Table 3: Comparison of concept selection methods across three benchmark datasets. Bold values highlight the highest accuracy (Accuracy in %).

| Method | CIFAR-10 | UCF101 | CUB200 |
|---|---|---|---|
| Random | 81.25 | 81.79 | 57.54 |
| Similarity | 83.79 | 80.85 | 52.08 |
| Orthogonality | 79.31 | 77.51 | 61.63 |
| Submodular | 87.18 | 83.29 | 59.71 |
| Lasso (Ours) | **92.26** | **85.44** | **65.79** |

Table 4: Ablation study of the model architecture. Accuracy is reported in %. Bold values indicate the accuracy of our model per dataset.

| Method Variant | CIFAR-10 | UCF101 | CUB200 |
|---|---|---|---|
| CFM (Ours) | **92.26** | **85.44** | **65.79** |
| Linear Probe | 94.75 | 94.52 | 72.40 |
| w/o Concept Matrix | 94.67 | 93.73 | 72.14 |
| w/o Path Calibration | 91.72 | 81.72 | 64.84 |
| w/o Node Temperature | 91.82 | 84.65 | 62.62 |
| + Hierarchical Loss | 91.13 | 83.78 | 65.38 |

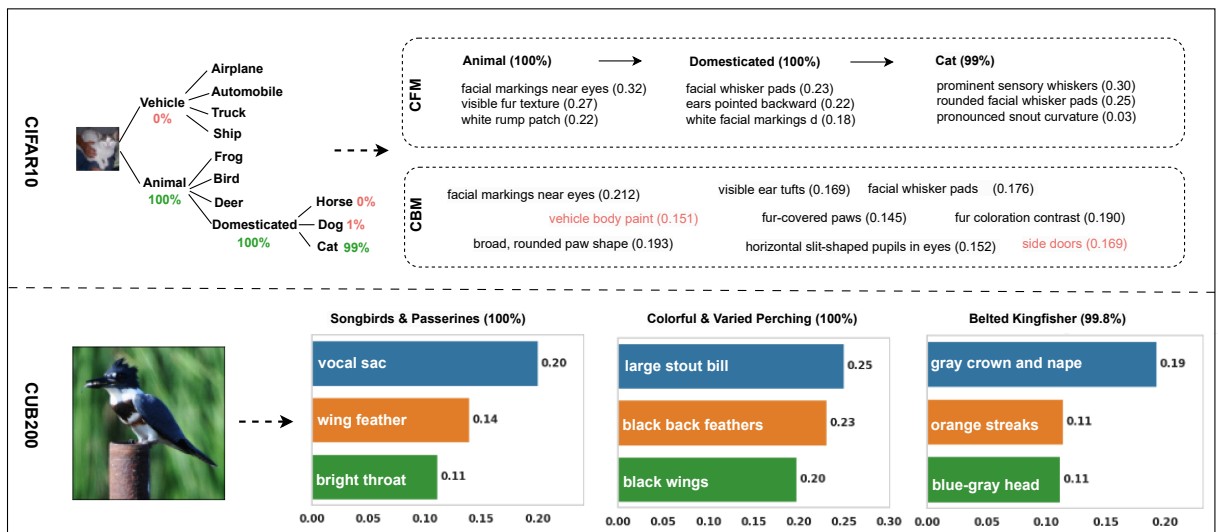

Figure 5: CFM decision path visualizations. **Top:** CFM and CBM explanations for a CIFAR-10 "cat" sample. **Bottom:** CFM decision path visualizations for selected samples from CUB-200. All models are trained using the CLIP-ViT-L/14 backbone, and candidate concepts are generated via GPT-4.1-mini. The top 3 activated concepts per node for CFM and the top 9 activated concepts for CBM are listed.

the image as an animal, then leverages "whisker pads" and "pointed ears" to rule out frogs, birds and deer, and finally relies on "sensory whiskers" and "rounded facial whisker pads" to distinguish cat from similar classes such as dog or horse. In contrast, CBM lacks such structured reasoning and treats all concepts as a flat set, which may occasionally lead to the misuse of irrelevant concepts for prediction. For instance, in the same cat example, CBM activates concepts like "vehicle body paint" and "side doors", which CFM assigns to the vehicle concept space and are unrelated to animals. This illustrates how CFM structurally isolates irrelevant concepts, thereby reducing information leakage.

The bottom part of Fig. 5 shows a CFM decision path example from CUB-200. The prediction follows the LLM-annotated route *Songbirds & Passerines → Colorful & Varied Perching → Belted Kingfisher*, with activated concepts at each node reflecting visual attributes of the image. (e.g., "vocal sac", "blue-gray head"). These stepwise explanations illustrate how CFM progressively narrows the prediction space, offering additional intuition and traceability for model decisions. However, CFM can still activate concepts not actually present in the image (e.g., "orange streaks" for this Belted Kingfisher), even when concept-generation prompts are enhanced with sample images from the training set. This phenomenon, also noted in prior work (Yang et al., 2023; Oikarinen et al., 2023), likely arises from CLIP's reliance on global image–text alignment without explicit region- or object-level grounding. Leveraging region-aware VLMs or grounded object detectors (Zhong

et al., 2022; Chen et al., 2024) may help mitigate this issue. We provide further qualitative analyses and extended examples in Appendix B.7.

### 5.6 Discussion

CFM reconciles accuracy and interpretability by enforcing structured concept utilization through semantically coherent, hierarchical bottlenecks. In this framework, each internal node is assigned a subset of discriminative concepts, which are only shared by the leaf classes under that node. This structural constraint localizes concept usage, ensuring that irrelevant concepts are isolated from class predictions outside the decision path and thereby effectively mitigating information leakage. As a result, CFM achieves a significantly lower number of effective concepts per prediction and higher semantic improvement ratios (SIR) compared to flat CBMs. By anchoring decisions at human-interpretable nodes, prioritizing task-specific concepts, and aggregating concept-weighted routing, CFM performs stepwise, semantically grounded inference.

**Broader impact.** Interpretable models should align reasoning processes with human cognitive frameworks. While traditional CBMs explain predictions through disconnected concept activations, CFM constructs human-aligned decision hierarchies from semantic embeddings, mirroring structural human reasoning patterns. By distributing discriminative concepts across LLM-annotated nodes (Sec. 3.2) and training interpretable hierarchical decision flows (Sec. 3.4), CFMs provide auditable sequential reasoning paths rather than isolated concept contributions. This structured approach achieves improved semantic utility over CBMs while matching their accuracy (Table 1), proving interpretability frameworks can adopt human-like reasoning hierarchies without sacrificing performance—a critical advancement for high-stakes applications requiring transparent decision-making.

**Limitations and mitigations.** Three challenges persist: 1) *Interpretability*: CFM's interpretability fundamentally relies on the quality of visual-language alignment provided by VLMs and the relevance of LLM-generated concepts. As discussed previously, some concepts are not faithfully grounded in the image and may reflect generic semantic knowledge rather than concrete visual evidence, which can limit the utility of explanations. This issue is exacerbated in specialized domains (e.g., medical or satellite imagery) that differ from web-crawled training data. This highlights the importance of backbone selection and candidate concept generation as prerequisites for both interpretability and accuracy. 2) *Fixed Tree Hierarchy*: CFM employs a fixed tree structure, assuming that class relationships are inherently hierarchical. While this is a common inductive bias in taxonomic classification, it is a strong assumption that may not hold for all tasks, particularly those lacking clear semantic hierarchies or requiring multiple overlapping category structures. Extending CFM to support alternative or dynamic relational structures could enhance its generality and applicability. 3) *Generalization*: While CFM leverages CLIP's vision-language alignment, its reliance on such backbones currently limits its generalizability. Integrating learnable projection layers (Oikarinen et al., 2023) may enable CFMs to be extended to non-vision-language architectures and other domains.

## 6 Conclusion

We proposed the Concept Flow Model (CFM), a hierarchical bottleneck architecture that enforces structural sparsity in concept-based models by restricting each prediction to a path-specific subset of concepts. Our results show that CFMs can match the accuracy of flat CBMs under the same concept budget, while substantially reducing the number of effective concepts per prediction and mitigating information leakage—particularly by limiting accuracy gains from random, task-irrelevant concepts and increasing reliance on meaningful semantic ones. These findings indicate that hierarchical bottlenecks may provide a promising direction for enhancing interpretability without sacrificing predictive performance. Future work will explore adaptive or dynamic hierarchical structures to increase flexibility and generality, investigate CFM's potential as structured context for VLMs to reduce hallucinations, and address limitations in visual groundedness for more reliable interpretability.

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

# A    Implementation Details

## A.1    Tree hierarchy extraction

**Tree generation.**    We implement multiple strategies to generate the decision tree structure $T$:

- **Data-induced tree:** We first compute class centroids by averaging CLIP image embeddings of training instances per class. Then, we apply hierarchical clustering (Ward's linkage (Ward Jr, 1963)) on these centroids to obtain the linkage matrix $Z$ that defines the merge structure.

- **Weights-induced tree:** Instead of computing class centroids from data, we use the learned linear classifier weights from a trained linear probe as the class representatives, and perform hierarchical clustering on them.

- **Random tree:** We generate entirely random vectors (matching the dimensionality of the class centroids) and perform hierarchical clustering on these vectors to create a purely random hierarchy.

After obtaining $Z$, we build the tree bottom-up. Each merge in $Z$ creates a new internal node by combining two child nodes. To avoid excessive merging of semantically distinct classes, we prune internal nodes whose merge distances fall below a pruning threshold $\tau$, defined as a specific percentile (e.g., the 30th percentile) of all distances in $Z$. This pruning step produces a multi-branch hierarchy with internal nodes representing semantically coherent groups. For controlled experiments comparing with other CBMs, we prune the tree hierarchy using a threshold $\tau$, set to 0.6 for CIFAR-10, 0.96 for CIFAR-100, 0.98 for CUB-200 and Tiny ImageNet, and 0.96 for UCF101, in order to produce a tree with exactly four internal nodes and a maximum depth of four.

The tree is implemented using a `TreeNode` structure, maintaining references to children, parent, node classes, and computed centroids. We ensure structural consistency by post-order traversal pruning and update sibling relationships and levels for downstream processing.

**Tree annotation.** We annotate internal nodes to improve semantic interpretability. This process involves:

- **Prompt construction:** For each non-leaf node, we collect the set of leaf class names in its subtree. We design a prompt that asks a large language model (LLM) to summarize the set into concise cluster names. The prompt format is as follows:

  ```
  You are an experienced taxonomist analyzing the dataset {dataset_name}.
  Given {num_clusters} clusters, each containing the following classes:
  {Cluster 0:  [class0, class1, ...], Cluster 1:  [...], ...}
  Your task is to identify common characteristics and assign a short, clear,
  exclusive name to each cluster.
  Return the result as JSON: { 'cluster_0':  'name0', 'cluster_1':  'name1', ...  }
  ```

- **LLM querying:** We use either GPT-4.1 from OpenAI (by default) or alternative LLMs (e.g., DeepSeek) to generate names. Responses are parsed to ensure valid JSON format and semantic consistency. We attempt multiple retries (set to 5 by default) if the output parsing fails.

- **Node naming:** Each internal node assigns its generated name to its immediate children, thereby incrementally annotating the hierarchy during breadth-first traversal.

## A.2 Candidate concept generation

Across the experiments, we generate three types of concepts: random concepts for non-semantic baselines, existing concepts from Labo (Yang et al., 2023) to enable controlled comparisons between CFM and prior CBMs, and LLM-generated concepts for visualization and interpretation of CFMs.

**Random concepts.** To establish a random baseline, we generate a large pool of purely random phrases independent of the dataset or hierarchy structure.

The random concept generation process proceeds as follows:

- **Word list collection:** We first download an English word list containing approximately 500,000 unique words from a public repository[1].

- **Phrase construction:** We randomly sample phrases by concatenating between 1 and 5 randomly selected words. Each phrase forms a simple noun-like structure (e.g., `river bright stone`, `glowing arc monument`), introducing linguistic randomness.

- **Storage:** We generate a fixed number (e.g., 30,000) of such random phrases and store them in a JSON file for reuse across experiments. Each phrase is indexed by a unique integer key.

- **Phrase sampling during training:** During training or evaluation, random phrases are drawn from the pre-generated pool to simulate random concept sets.

---

[1] `https://raw.githubusercontent.com/dwyl/english-words/master/words.txt`

This random phrase generation pipeline provides a controlled method to assess the model's robustness against semantically meaningless features, serving as a baseline when evaluating the effectiveness of selected concepts from the existing or LLM-generated concept pool.

**Existing concepts.** We reuse pre-existing LLM-generated concepts to ensure a fair baseline model comparison. Specifically, we leverage the concepts provided by LaBo (Yang et al., 2023), which are publicly available[2].

The concept loading process proceeds as follows:

- **Concept file retrieval:** For each dataset, we access the corresponding JSON file containing class-to-concept mappings, where each class is associated with a list of LLM-generated visual concepts.

- **Concept extraction:** For each internal node, we iterate through its leaf classes and collect all concepts associated with the classes from the pre-loaded mappings.

This procedure enables experiments using high-quality, curated concept sets without requiring costly LLM queries.

**LLM-generated concepts** We generate candidate concepts *per leaf class* and only aggregate them to internal nodes afterwards. Concretely, we first run a per-class concept discovery stage and then construct node-level candidate pools by unioning concepts from descendant leaves.

- **Per-class discovery (multimodal prompts).** For each leaf class $c$, we sample up to `max_images_per_class` training images and group them into batches of size `images_per_prompt` (discarding groups smaller than `min_images_per_group`). Each request to the LLM attaches the image batch (as base64-encoded PNGs) *before* a textual prompt that includes: the target class name, its parent-chain, its sibling classes, and the set of concepts already discovered for $c$ (to discourage repetition). The prompt also enforces constraints on the number of parent-consistent and sibling-distinguishing traits and a maximum phrase length.

- **LLM querying and parsing.** We query an OpenAI chat model (default: GPT-4.1-mini, configurable via `model_name`) with constant backoff and at most `max_retry_attempts`. The model returns short noun phrases; we parse the response line-wise, strip numbering, enforce a `max_words_per_concept` limit, and keep only novel items relative to the per-class memory set. We iterate image batches until reaching `max_concepts_per_class` or exhausting data, with an `api_call_delay` between calls.

- **Caching and resumption.** Concepts discovered for each class are saved incrementally to a JSON file (`<dataset>_<backbone>_llm_concepts.json`). When `resume` is enabled, previously completed classes are skipped and generation continues for the remaining classes.

- **Bottom-up aggregation at internal nodes.** After per-class discovery, we traverse internal nodes in a bottom-up order and construct each node's candidate pool by aggregating the concepts of its descendant leaf classes. We also retain the originating class index for each concept to preserve the linkage between concepts and classes. No additional LLM calls are made at this stage.

This two-stage design grounds concepts in real images for each class while leveraging hierarchical context (parents and siblings) to promote class-specificity and sibling contrast. The subsequent bottom-up aggregation yields node-local candidate pools without extra LLM cost, supporting fine-grained classification and interpretable, hierarchy-aware decisions.

## A.3 Concept preprocessing

Before using LLM-generated candidate concepts for concept selection, we apply two preprocessing steps to ensure quality and semantic diversity:

---

[2]`https://github.com/YueYANG1996/LaBo/tree/main/datasets`

**Removal of concepts containing class names.** To prevent data leakage and trivial shortcuts, we filter out candidate concepts that contain any class names or substrings thereof. For each candidate concept $c$, we perform a case-insensitive regular expression search against all class names. Concepts matching any class name pattern are discarded.

**Removal of semantically redundant concepts.** To encourage concept diversity, we eliminate highly similar concepts within each node. Specifically, we:

- Encode all candidate concepts into CLIP text embeddings using the backbone model.

- Normalize all embeddings and compute pairwise cosine similarities.

- For each concept, we remove subsequent concepts whose cosine similarity exceeds a predefined threshold $\theta$ (We set $\theta = 0.9$ for all experiments).

This redundancy removal is performed in a greedy, batch-efficient manner to minimize memory usage. We prioritize keeping earlier concepts when conflicts arise and explicitly free unused tensors to reduce CPU memory overhead.

Overall, this two-stage cleaning process ensures that the remaining candidate concepts for each internal node are both distinct and informative, leading to better semantic coverage and improving interpretability in later concept selection stages.

### A.4 Concept distribution

To determine the concept budget $r_v$ for each internal node $v$, we employ a hybrid allocation strategy that combines a fixed base share with an adjustment based on the node's child count:

$$r_v = \max\big(\alpha \, |\text{Child}(v)| + (1-\alpha) \, |B|, \, r_{\min}\big), \qquad v \in \mathcal{V}_{\text{int}}. \tag{11}$$

Here, $B = \frac{N}{m}$ denotes the mean (base) share when $N$ concepts are distributed across $m$ internal nodes. The coefficients $\alpha \in [0,1]$ and $(1-\alpha)$ balance the influence of the child-weighted term and the uniform baseline, respectively. $r_{\min}$ enforces a minimum number of concepts allocated per node. Further implementation details are provided in the supplementary code.

**Hyper-parameters.**

- $\alpha$: weight of the child-count term (default 0.2).
- $r_{\min}$: minimum number of concepts per node (default 1).

We fix $r_{\min} = 1$ for all experiments. The child-weighted coefficient $\alpha$ is tuned from 0 to 1 in increments of 0.1. The final settings are chosen as follows: $\alpha = 0.6$ for CIFAR-10, CIFAR-100, and Tiny ImageNet; $\alpha = 0.9$ for UCF-101; and $\alpha = 0.8$ for CUB-200.

### A.5 Concept selection

After generating and preprocessing candidate concepts for each internal node, we apply a concept selection step to choose a compact subset of informative concepts. The selection process is organized as follows:

**Multi-class data preparation.** For each internal node, we collect training examples corresponding to the child classes of that node. We construct a multi-class prediction task where the goal is to predict the child class from instance features, enabling quantitative evaluation of concept quality.

**Selection strategies.** We implement five different strategies to select a subset of concepts:

- **Lasso selection:** We represent each sample as a vector of cosine similarities to candidate concept embeddings. We then apply $\ell_1$-penalized logistic regression (Lasso) with the following settings for all experiments: `penalty='l1'`, `solver='liblinear'`, `multi_class='ovr'`, and `max_iter=1000`. The resulting sparse model identifies a compact set of concepts that are predictive of child classes. We select the top-$r$ ranked concepts, where $r$ is the desired number of concepts per node.

- **Similarity selection:** For each concept, we compute its average cosine similarity to all training samples. Concepts with the highest average similarity scores are selected.

- **Orthogonality selection:** We use a facility location-based submodular optimization approach to select the top-$r$ concepts that are maximally diverse (i.e., mutually orthogonal in embedding space).

- **Random selection:** Concepts are randomly shuffled and the top-$r$ concepts are selected as a baseline.

- **Submodular selection:** We use a mutual information-augmented submodular optimization method, balancing concept relevance and diversity. This approach follows the procedure outlined in LaBo (Yang et al., 2023) and operates by selecting concepts that maximize a mixture of informativeness and diversity scores.

## A.6  Hierarchical loss

We add a hierarchical auxiliary loss to supervise routing decisions at internal nodes of the tree. At each internal node, the model predicts a probability distribution over its children based on the input embedding. We apply a cross-entropy loss between the predicted distribution and the ground-truth child node, determined by the path leading to the correct leaf class. The hierarchical loss is accumulated along the path from the root to the leaf and is added to the main classification loss during training. This auxiliary supervision encourages more accurate intermediate decisions but can slightly degrade overall performance, consistent with observations from prior work (Wan et al., 2020).

# B  Experiment details

## B.1  Dataset statistics

Table 5 summarizes the key statistics of the datasets used in our experiments, including the number of classes and the number of samples in the training, validation, and test sets. For datasets without predefined validation splits, we randomly split the training set into 80% for training and 20% for validation.

Table 5: Dataset statistics.

| Dataset | # Classes | Train | Validation | Test |
|---|---|---|---|---|
| CIFAR-10 | 10 | 40,000 | 10,000 | 10,000 |
| CIFAR-100 | 100 | 40,000 | 10,000 | 10,000 |
| CUB-200-2011 | 200 | 4,795 | 1,199 | 5,794 |
| Tiny ImageNet | 200 | 80,000 | 20,000 | 10,000 |
| UCF101 (processed) | 101 | 7,639 | 1,898 | 3,783 |

- **CIFAR-10** and **CIFAR-100** have predefined 50,000 training and 10,000 test images. We split the original training set into 80% training and 20% validation.

- **CUB-200-2011** consists of 11,788 images across 200 bird species, with a standard split of 5,994 training and 5,794 test images. We split the training set into 80% training and 20% validation.

- **Tiny ImageNet** provides 100,000 training images and 10,000 test images across 200 classes. We split the original training set into 80% training and 20% validation.

- **UCF101** refers to the processed version released by Yang et al. (2023) based on mid-frames from videos, available at `https://drive.google.com/uc?id=10Jqome3vtUA2keJkNanAiFpgbyC9Hc2O`. We split the training set into 80% training and 20% validation.

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

## B.2  Experiment settings

Our experiments are conducted on a system equipped with Intel Xeon E5-2640 v4 CPUs (20 cores, 40 threads) operating at 2.40 GHz, 503 GiB of RAM, and an NVIDIA RTX A6000 GPU with approximately 49 GiB of VRAM. All data are processed using the CLIP ViT-B/32 (Radford et al., 2021) backbone (unless otherwise specified) and cached to accelerate model training. Two worker threads are used for data processing. For all models, we train only the linear layers based on the cached, featurized data. Constructing a Concept Flow Model, including hierarchy generation, concept selection, and training, takes between 0.1 and 5 hours depending on the dataset. All hyperparameter settings are provided in Sec. A, with additional details for Linear Probe, PCBM, and Labo in the supplementary code and YAML configuration files.We do not explicitly include NBDT(Wan et al., 2020) as a baseline for comparison, as it was not designed to support concept-level explanations. However, our ablation study in Sec. 5.4, which removes the concept matrix, results in a model that is approximately equivalent to NBDT and achieves performance comparable to the non-interpretable linear probe baseline.

**Training setup.** All models are trained using the Adam optimizer with an initial learning rate of 0.01. We use a mini-batch size of 128 (unless otherwise specified) and train for up to 200 epochs with early stopping based on validation loss. If the validation loss does not improve by at least 0.0001 for 5 consecutive epochs, training is terminated early. The learning rate is reduced by a factor of 0.5 if the validation loss plateaus for 3 epochs, following a ReduceLROnPlateau schedule.

All experiments are implemented in PyTorch and executed on CUDA-enabled devices. Random seeds are fixed across data splits, concept processing and model initialization to ensure reproducibility. Concept generation, selection, training, and evaluation pipelines are fully automated and configurable via YAML configuration files.

### B.3 Experiment with Elastic Net regularization

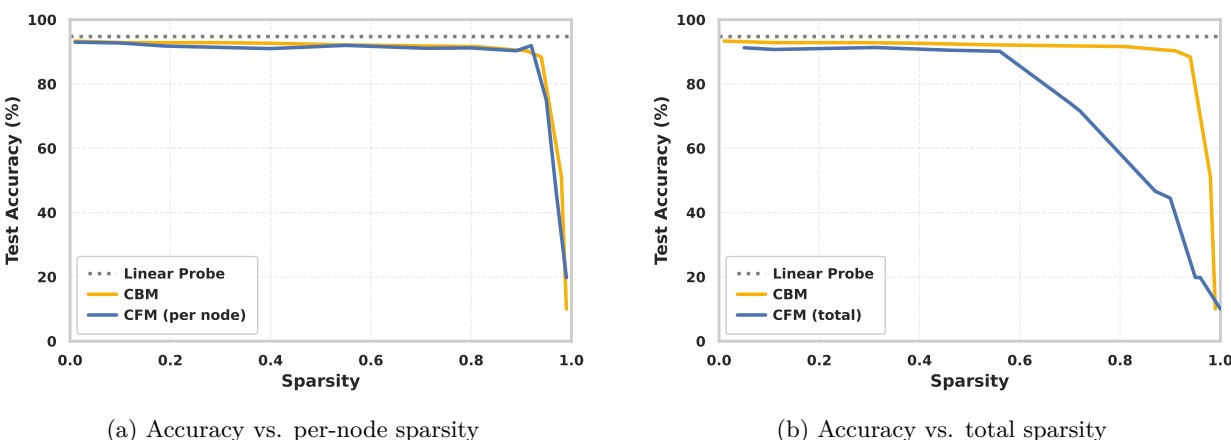

(a) Accuracy vs. per-node sparsity          (b) Accuracy vs. total sparsity

Figure 6: **Impact of Elastic Net regularization-induced sparsity on accuracy.** (Left) Accuracy trends for the first experimental group, where CFM (4000 total concepts, 1000 per node) is initialized with four times more random concepts than CBM (1000 total concepts). As the number of effective concepts per node (CFM) or layer (CBM) is progressively reduced, both models exhibit similar accuracy trends. (Right) Accuracy trends for the second experimental group, where CBM and CFM are initialized with the same total number of random concepts (4000).

Previous works (Srivastava et al., 2024; Yuksekgonul et al., 2022) have applied Elastic Net regularization (Zou & Hastie, 2005)—which combines $\ell_1$ and $\ell_2$ penalties—to the concept-to-class linear layer in concept-based models to control the number of effective concepts and improve robustness. However, as noted by Srivastava et al. (2024), tuning the regularization strength to target a specific number of effective concepts can be labor-intensive and dataset-dependent. For this reason, we omit Elastic Net regularization in our controlled experiments (Sec. 5.2) and instead directly control the number of concepts.

Nonetheless, we conduct additional experiments here to validate that Elastic Net regularization can achieve similar effects. Specifically, we train CBMs and CFMs on CIFAR-10 using purely random concepts, applying Elastic Net regularization to induce the sparsity of the linear layers.

Following Yuksekgonul et al. (2022), we define the regularization objective as:

$$\frac{\lambda}{N_c K}\Omega(\mathbf{w}),$$

where $N_c$ is the number of concepts, $K$ is the number of classes, $\lambda$ controls regularization strength, and $\Omega(\mathbf{w})$ is the Elastic Net penalty:

$$\Omega(\mathbf{w}) = \alpha\|\mathbf{w}\|_1 + (1-\alpha)\|\mathbf{w}\|_2^2.$$

We fix the mixing ratio $\alpha = 0.9$ to emphasize sparsity through $\ell_1$ regularization. We consider two experimental groups: (1) CBM is initialized with 1000 concepts, and CFM is initialized with 1000 concepts per internal

node (resulting in 4000 total concepts across 4 nodes); (2) CBM is initialized with 4000 concepts, while CFM maintains 1000 concepts per node (4000 total concepts). In both setups, we measure sparsity as the proportion of near-zero weights ($|w_i| < 10^{-2}$) in the concept weight matrices. We vary the regularization strength $\lambda$ across 20 logarithmically spaced values between $\lambda_{\min} = 0.01$ and $\lambda_{\max} = 20.0$ to achieve a range of target sparsity levels for comparison.

Figure 6 (left) shows the results for the first experimental group. As the number of effective concepts decreases (i.e., sparsity increases), the accuracy of both CBM and CFM declines in a similar manner, mirroring the trends observed when directly reducing/increasing the number of concepts (see Fig. 3). In contrast, Figure 6 (right) presents the second experimental group, where CBM and CFM are initialized with the same total number of concepts (4000). In this setting, CFM experiences an earlier and sharper accuracy drop as sparsity increases. This is because the smaller number of random concepts per node in CFM limits each node's discriminative capacity, making it harder to maintain high classification accuracy compared to CBM.

For convenience, we omit Elastic Net regularization in the main paper's controlled experiments to enable precise control over concept count and facilitate comparison with Labo and PCBM. However, in practical applications, we recommend using Elastic Net regularization in CFM, particularly when a node contains a large number of concepts (e.g., CUB dataset with 100 concepts per node).

## B.4 Detailed Experimental Results for Scenario 1

Table 6: Detailed results for Figure 3. All values are accuracy (%) reported as mean $\pm$ std over three runs.

(a) Fixed total budget $R$

| $R$ | CFM | CBM |
|---|---|---|
| 8 | $30.11 \pm 3.77$ | $63.46 \pm 3.95$ |
| 16 | $41.90 \pm 5.15$ | $72.02 \pm 2.01$ |
| 32 | $60.76 \pm 0.48$ | $84.01 \pm 0.55$ |
| 64 | $75.95 \pm 3.68$ | $89.78 \pm 0.26$ |
| 128 | $86.30 \pm 0.78$ | $91.26 \pm 0.10$ |
| 256 | $88.94 \pm 1.74$ | $92.25 \pm 0.12$ |
| 512 | $90.39 \pm 1.41$ | $92.63 \pm 0.06$ |

(b) Fixed bottleneck concepts

| $R$ | CFM | CBM |
|---|---|---|
| 2 | $30.11 \pm 3.77$ | $30.79 \pm 2.98$ |
| 4 | $41.90 \pm 5.15$ | $42.20 \pm 4.98$ |
| 8 | $60.76 \pm 0.48$ | $56.14 \pm 0.06$ |
| 16 | $75.95 \pm 3.68$ | $76.86 \pm 2.73$ |
| 32 | $86.30 \pm 0.78$ | $85.49 \pm 0.86$ |
| 64 | $88.94 \pm 1.74$ | $89.68 \pm 0.06$ |
| 128 | $90.39 \pm 1.41$ | $91.04 \pm 0.07$ |
| 256 | $90.63 \pm 1.86$ | $92.25 \pm 0.12$ |
| 512 | $90.33 \pm 1.51$ | $92.63 \pm 0.06$ |

(c) CFM pruning ($R = 60$)

| Inner Nodes | Accuracy |
|---|---|
| 1 | $88.41 \pm 0.78$ |
| 2 | $85.95 \pm 0.88$ |
| 3 | $80.76 \pm 1.41$ |
| 4 | $71.86 \pm 3.96$ |
| 5 | $71.76 \pm 3.30$ |
| 6 | $67.33 \pm 4.03$ |
| 7 | $61.97 \pm 1.75$ |
| 8 | $61.88 \pm 2.65$ |
| 9 | $54.05 \pm 1.56$ |

(d) CBM sparsity ($R = 60$)

| $\lambda$ | Accuracy |
|---|---|
| 0.1 | $84.53 \pm 0.90$ |
| 0.2 | $81.37 \pm 1.46$ |
| 0.3 | $75.84 \pm 1.33$ |
| 0.4 | $74.00 \pm 0.29$ |
| 0.5 | $70.02 \pm 1.31$ |
| 0.6 | $56.36 \pm 13.07$ |
| 0.7 | $56.02 \pm 9.35$ |
| 0.8 | $37.29 \pm 4.73$ |
| 0.9 | $38.75 \pm 9.42$ |

## B.5 Detailed Experimental Results for Scenario 2

Table 7: Full experimental results with standard deviations over 3 runs. NEC: number of effective concepts; Acc: classification accuracy (%); SIR: semantic improvement over random concepts (%).

| Dataset | Method | NEC | Acc (%) | SIR (%) |
|---|---|---|---|---|
| CIFAR-10 | Linear Probe | N/A | 94.59±0.16 | N/A |
| | CBM (Random) | 56.80±0.82 | 89.20±1.18 | 0 |
| | CFM (Random) | 6.90±0.00 | 76.46±0.82 | 0 |
| | PCBM | 56.80±0.52 | 91.15±0.10 | 2.19 |
| | Labo | 57.73±0.15 | 90.93±0.12 | 1.94 |
| | CFM (Ours) | **6.90±0.10** | **91.59±0.38** | **19.78** |
| CIFAR-100 | Linear Probe | N/A | 77.53±0.34 | N/A |
| | CBM (Random) | 184.20±0.72 | 68.50±1.32 | 0 |
| | CFM (Random) | 54.24±10.12 | 62.00±1.11 | 0 |
| | PCBM | 183.99±0.13 | 68.98±1.69 | 0.70 |
| | Labo | 184.33±0.51 | 70.98±0.67 | 3.62 |
| | CFM (Ours) | **55.02±8.36** | **72.29±0.02** | **16.61** |
| UCF101 | Linear Probe | N/A | 95.00±0.38 | N/A |
| | CBM (Random) | 187.31±1.90 | 79.15±1.38 | 0 |
| | CFM (Random) | 44.99±1.79 | 72.76±2.23 | 0 |
| | PCBM | N/A | N/A | N/A |
| | Labo | 190.60±0.84 | **84.62±0.24** | 6.91 |
| | CFM (Ours) | **45.80±1.25** | 84.60±0.21 | **16.27** |
| CUB200 | Linear Probe | N/A | 71.68±0.60 | N/A |
| | CBM (Random) | 344.19±8.49 | 52.23±2.00 | 0 |
| | CFM (Random) | 104.61±12.47 | 47.08±2.22 | 0 |
| | PCBM | N/A | N/A | N/A |
| | Labo | 357.92±0.77 | 65.13±0.34 | 24.70 |
| | CFM (Ours) | **109.34±13.13** | **65.68±0.51** | **39.50** |
| TinyImageNet | Linear Probe | N/A | 75.25±0.16 | N/A |
| | CBM (Random) | 362.34±3.24 | 68.78±0.71 | 0 |
| | CFM (Random) | 89.83±0.16 | 64.89±0.59 | 0 |
| | PCBM | 362.01±0.43 | 71.07±0.31 | 3.32 |
| | Labo | 369.36±0.57 | 69.86±0.34 | 1.57 |
| | CFM (Ours) | **92.12±4.71** | **70.79±0.28** | **9.09** |

## B.6 Comparison of CFM and PCBM with Sparsity Regularization

We conduct additional experiments comparing CFM and PCBM with and without sparsity regularization. While the main experiments in Section 5 disable sparsity regularization to isolate the effect of CFM's structural design from regularization-induced sparsity, we demonstrate here that CFM's advantages persist, when sparsity regularization is applied to both methods.

**Experimental Setup.** We compare PCBM and CFM on CIFAR-10 under two conditions: (1) without sparsity regularization, (2) with sparsity regularization at matched NEC levels of 5 and 3. Following the original PCBM implementation (Yuksekgonul et al., 2022), we set the L1 ratio to 0.99 and tune the regularization strength $\lambda$ to achieve the target NEC values. This methodology follows VLG-CBM (Srivastava et al., 2024), which emphasizes that fair comparison requires matching effective concept counts across methods.

**Results.** Table 8 presents the results. Without sparsity regularization, CFM achieves comparable accuracy to PCBM (91.68% vs. 91.36%) while using substantially fewer effective concepts (7 vs. 57). When sparsity regularization is applied to match NEC values, CFM significantly outperforms PCBM: at NEC=5, CFM

achieves 90.66% accuracy compared to PCBM's 85.85% (+4.81%); at NEC=3, CFM maintains 90.61% accuracy while PCBM drops to 84.41% (+6.20%).

Table 8: Comparison of PCBM and CFM with and without sparsity regularization on CIFAR-10. For experiments with regularization, $\lambda$ is tuned to achieve matched NEC values. Results demonstrate that CFM's structural sparsity complements sparsity regularization effectively.

| Method | Sparsity Reg. | NEC | Accuracy (%) | $\lambda$ |
|--------|:-------------:|:---:|:------------:|:---------:|
| PCBM | ✗ | 57 | 91.36 | — |
| CFM | ✗ | 7 | 91.68 | — |
| PCBM | ✓ | 5 | 85.85 | 0.8 |
| CFM | ✓ | 5 | 90.66 | 0.02 |
| PCBM | ✓ | 3 | 84.41 | 1.3 |
| CFM | ✓ | 3 | 90.61 | 0.1 |

These results demonstrate three key findings: (1) without sparsity regularization, CFM achieves comparable accuracy with substantially fewer effective concepts, validating Proposition 4.2; (2) with sparsity regularization at matched NEC levels, CFM significantly outperforms PCBM, indicating that structural sparsity provides benefits beyond what regularization alone can achieve; and (3) CFM's structural sparsity and sparsity regularization can be effectively combined. CFM requires much smaller $\lambda$ values to achieve the same NEC, suggesting that its hierarchical structure already enforces substantial concept isolation.

### B.7 Qualitative Analysis of Decision Paths

To complement the quantitative evaluation in Section 5, we present qualitative analyses of CFM decision paths. We select four samples across three datasets to illustrate both successful and unsuccessful predictions: three correctly classified cases demonstrating different levels of concept-image correspondence, and one failure case to examine CFM's diagnostic capabilities. All models use the CLIP ViT-L/14 backbone with candidate concepts generated by GPT-4.1-mini.

**Case-by-Case Observations.** Table 9 illustrates how CFM produces traceable decision paths, where each prediction follows a unique root-to-leaf trajectory through localized concept subsets.

*(a) Bowling.* With only two transitions, the model routes from "Root" to "Sports and Athletic Events" (99.99%) using motion-related concepts such as "ball trajectory on lane" (0.25), then to "bowling" (99.91%) via task-specific concepts including "ten-pin setup" (0.30) and "bowler's side-on stance" (0.21). However, "colored cue ball markings" (0.16)—which refers to billiards rather than bowling—illustrates that not all activated concepts are semantically correct, even in successful predictions.

*(b) Painted Bunting.* The three-level hierarchy progressively refines the prediction: generic bird features ("songbird vocal sac presence", "songbird wing feather pattern") at the first level, visual characteristics ("large stout bill", "glossy black back feathers") at the second, and species-level attributes at the leaf. The final transition activates two concepts, i.e., "iridescent green wing coverts" and "iridescent golden-green wing coverts", both referencing green coloration. While painted buntings do have greenish back feathers, the most visually prominent features in Figure 7b (blue head, red breast) are not captured by the selected concepts.

*(c) Parking Meter (Failure Case).* This misclassification illustrates how CFM's decision path can expose prediction uncertainty. At the second transition, the model shows only 69.24% confidence for "Everyday Objects & Apparel" versus 30.76% for "Structures & Vehicles", which is a notably uncertain split. The concepts activated throughout the path ("rows of stop knobs", "tactical utility belt", "graduated fluid chamber") show little semantic correspondence to the input image. The resulting path probability of 68.69% is clearly lower than the correctly classified samples before. TinyImageNet's low resolution (64×64) may further limit CLIP's ability to extract discriminative visual features.

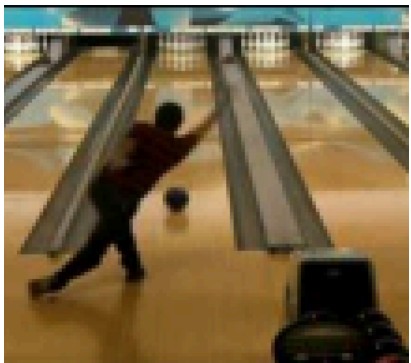

(a) Bowling (UCF-101): Correct

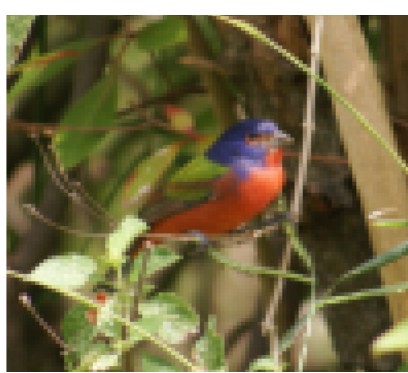

(b) Painted Bunting (CUB-200): Correct

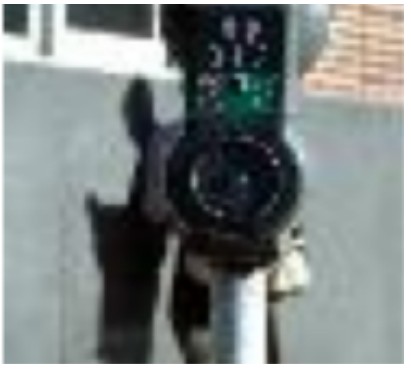

(c) Reel (TinyImageNet): Misclassified as Parking Meter

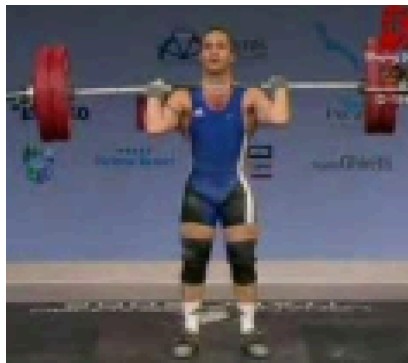

(d) Clean and Jerk (UCF-101): Correct

Figure 7: **Input samples for qualitative analysis.** We analyze CFM decision paths for four cases across three datasets. Detailed decision paths with concept activations are provided in Table 9.

*(d) Clean and Jerk.* The decision path transitions through "Personal Actions and Instrument Play" → "Physical Activities" → "clean and jerk". Upper-level concepts exhibit grounding issues: "standing musician posture" and "child-sized cleaning tool" do not correspond to the image. The latter may arise from spurious linguistic correlation with "clean" in the class name. In contrast, leaf-level concepts such as "weight plates visible on barbell" (0.28) and "weightlifting platform mat" (0.19) directly match salient visual elements. In this case, accurate prediction occurred despite poorly aligned upper-level concepts, suggesting that leaf-level concept quality may be more critical for final classification.

**Observations on Interpretability.** These examples highlight both the utility and limitations of CFM's decision paths.

*Traceability.* CFM's stepwise structure makes prediction reasoning explicit. In the parking meter failure case, the uncertain routing at the second transition (69.24% vs. 30.76%) and the semantically misaligned concepts provide interpretable signals of an unreliable prediction. This contrasts with flat CBMs, where all concepts contribute to a single linear combination, making it harder to identify which semantic distinctions caused an error.

*Concept alignment limitations.* Even in correctly classified samples, some activated concepts are semantically incorrect ("colored cue ball markings" for bowling) or fail to capture visually salient features (green wing coverts for a bird with prominent blue and red coloring). These observations align with limitations of

Table 9: **Decision paths and concept activations for samples in Figure 7.** Each row shows a transition, the branch probability, and the top activated concepts with weights (up to 3 per transition). Unlike flat CBMs that expose all concepts to every class prediction, CFM restricts each transition to a localized concept subset. Path Prob. denotes the cumulative product of transition probabilities.

| Sample | Transition | Prob. | Top Activated Concepts (weight) |
|---|---|---|---|
| (a) Bowling (UCF-101) Path Prob: 99.90% | Root → Sports and Athletic Events | 99.99% | ball trajectory on lane (0.25) spherical ball kicked by player (0.19) rider's legs bent at knees (0.18) |
| | Sports and Athletic Events → bowling | 99.91% | ten-pin setup (0.30) bowler's side-on stance (0.21) colored cue ball markings (0.16) |
| (b) Painted Bunting (CUB-200) Path Prob: 98.96% | Root → Songbirds & Passerines | 100.00% | songbird vocal sac presence (0.20) songbird wing feather pattern (0.14) bright chestnut throat (0.14) |
| | Songbirds & Passerines → Colorful and Varied Perching Birds | 98.99% | large stout bill (0.22) glossy black back feathers (0.21) glossy black wings and back (0.18) |
| | Colorful and Varied Perching Birds → painted bunting | 99.97% | iridescent green wing coverts (0.16) iridescent golden-green wing coverts (0.16) |
| (c) Parking Meter (TinyImageNet) Pred: reel Path Prob: 68.69% | Root → Objects & Structures | 100.00% | rows of stop knobs (0.41) series of tall vertical piers (0.39) waist-level fit (0.36) |
| | Objects & Structures → Everyday Objects & Apparel (vs. Structures & Vehicles 30.76%) | **69.24%** | tactical utility belt (0.22) graduated fluid chamber (0.18) side-tied bottom strings (0.17) |
| | Everyday Objects & Apparel → reel | 99.21% | designed for manual operation (0.43) stick extending lengthwise (0.31) tapered cooking depth (0.24) |
| (d) Clean and Jerk (UCF-101) Path Prob: 98.77% | Root → Personal Actions and Instrument Play | 98.85% | standing musician posture (0.16) padded weightlifting bench support (0.15) child-sized cleaning tool (0.15) |
| | Personal Actions and Instrument Play → Physical Activities | 100.00% | waist-level prop manipulation (0.15) climbing-specific grip technique (0.15) climbing shoes with rubber soles (0.14) |
| | Physical Activities → clean and jerk | 99.92% | weight plates visible on barbell (0.28) weightlifting platform mat (0.19) weightlifting barbell with weighted plates (0.18) |

CLIP-based concept alignment noted in prior work (Oikarinen et al., 2023; Yang et al., 2023), indicating that concept pool quality remains important for explanation fidelity regardless of the model architecture.

### B.8 ImageNet-1K Experiments

To validate the scalability of CFM to large-scale datasets, we conduct experiments on ImageNet-1K, which contains 1000 classes and over 1.2 million training images.

**Scalability via Stratified Sampling.** The computational bottleneck arises from computing the similarity matrix $S_v \in \mathbb{R}^{n \times n_v}$ and solving Lasso regression over all $n$ samples during concept selection. For ImageNet-1K, processing all training samples is prohibitive. We address this via stratified sampling: grouping samples by class and drawing a balanced subset with $n_{max} = 15{,}000$ samples. Since concept selection requires representative samples rather than exhaustive coverage, this preserves class structure while remaining tractable regardless of dataset size.

**Experimental Setup.** We use CLIP ViT-B/32 as the backbone (512-dimensional embeddings) with a total budget of 1000 concepts (one per class). CFM is configured with 5 internal nodes and 3 levels. All other settings follow Section 5.1. We report mean and standard deviation over 3 runs with different random seeds.

**Results.** Table 10 summarizes the results. CFM achieves competitive accuracy (72.88%) with significantly fewer effective concepts (339 vs. ∼860). With 1000 concepts exceeding the 512-dimensional embedding, flat CBMs exhibit severe information leakage. CBM (Random), PCBM, and Labo all fall within 0.25% of Linear Probe, with near-zero SIR confirming that random concepts suffice when the concept count exceeds the embedding dimension. In contrast, CFM limits per-node concepts below the embedding dimension through its hierarchical structure, yielding substantially higher SIR (4.70%) and validating hierarchical bottlenecks' resistance to information leakage at scale.

Table 10: Comparison of methods on ImageNet-1K. We report the number of effective concepts (NEC), classification accuracy (Acc), and semantic improvement over random concepts (SIR). Results are mean ± std over 3 runs. Bold values indicate best performance among concept-based methods.

| Method | NEC | Acc (%) | SIR (%) |
|---|---|---|---|
| Linear Probe | N/A | $73.03 \pm 0.63$ | N/A |
| CBM (Random) | $853.92 \pm 16.70$ | $72.82 \pm 0.43$ | 0 |
| CFM (Random) | $260.10 \pm 1.62$ | $69.61 \pm 0.23$ | 0 |
| PCBM | $860.44 \pm 8.18$ | $73.08 \pm 0.76$ | $0.37 \pm 1.51$ |
| Labo | $868.31 \pm 0.69$ | $72.86 \pm 0.55$ | $0.06 \pm 0.42$ |
| CFM (Ours) | $\mathbf{339.18 \pm 1.12}$ | $\mathbf{72.88 \pm 0.56}$ | $\mathbf{4.70 \pm 0.58}$ |

## C  Theoretical Analysis

### C.1  Computational Complexity Analysis

We compare CBMs and CFMs under the same concept budget $R$ and we assume modern GPUs can parallelize independent nodes at the same tree level; different levels are computed sequentially (root $\to$ leaves).

**Notation.** $d$: CLIP embedding dimension; $k$: number of classes (leaf nodes); $n$: number of training samples; $R$: total number of concepts; $m$: number of internal nodes in CFM; $b$: average branching factor; $b_{\max}$: maximum branching factor; $\ell$: average root-to-leaf path length; $r' = R/m$: average number of concepts per internal node. In a rooted tree with $k$ leaves and $m$ internal nodes, the sum of the numbers of children over all nodes equals the total number of edges, which is $k + m - 1$.

#### C.1.1  Forward Pass Complexity

**CBM Complexity.**

- Concept projection: $\tilde{z}\tilde{C}^\top$ with $C \in \mathbb{R}^{R \times d}$: $\mathcal{O}(Rd)$

- Classification: $sW$ with $W \in \mathbb{R}^{R \times k}$: $\mathcal{O}(Rk)$

- Total: $\mathcal{O}(Rd + Rk)$

**CFM Sequential Complexity.**

- Concept projections across all internal nodes: $\sum_v |C_v| \cdot d = Rd \Rightarrow \mathcal{O}(Rd)$

- Local branching multiplications: $\sum_v r' \cdot |\text{Ch}(v)| = \frac{R}{m}(k + m - 1) \Rightarrow \mathcal{O}\left(R + \frac{Rk}{m}\right)$

- Node-wise softmax + probability propagation: touch each edge once $\Rightarrow$ $\mathcal{O}(k+m)$

- Total: $\mathcal{O}\big(Rd + \frac{Rk}{m} + R + k + m\big) \approx \mathcal{O}\big(Rd + \frac{Rk}{m} + k + m\big)$

**CFM Parallel Complexity (level-wise parallelism).** Assuming perfect parallelization across nodes at the same depth, the per-level cost is $\mathcal{O}\big(\frac{Rd}{m} + \frac{Rb}{m}\big)$ (concept projections plus local branching). Over $\ell$ levels and assembling leaf scores:

$$\mathcal{O}\Big(\ell\big(\tfrac{Rd}{m} + \tfrac{Rb}{m}\big)\Big) + \mathcal{O}(\ell b_{\max}).$$

For $d \gg b$, this is well-approximated by $\mathcal{O}\big(\ell \frac{Rd}{m}\big)$.

### C.1.2 Comparison Under Different Scenarios

**Inference (all-class logits).** It is standard to compute all class logits once and take $\arg\max$.

- CBM: $\mathcal{O}(Rd + Rk)$

- CFM (parallel): $\mathcal{O}\Big(\ell\big(\tfrac{Rd}{m} + \tfrac{Rb}{m}\big)\Big) + \mathcal{O}(\ell b_{\max})$

**Training (negative log-likelihood).** With loss $-\log p_{v_0 \to y}(x)$, CFM only evaluates the *unique* path to the ground-truth leaf $y$:

$$\text{CFM (per sample)}: \ \mathcal{O}\Big(\ell\big(\tfrac{Rd}{m} + \tfrac{Rb}{m}\big)\Big) \ + \ \mathcal{O}(\ell b) \text{ (lower-order)}, \qquad \text{CFM (dataset)}: \ \mathcal{O}\Big(n\,\ell\big(\tfrac{Rd}{m} + \tfrac{Rb}{m}\big)\Big).$$

CBM (all-class logits) remains $\mathcal{O}\big(n(Rd + Rk)\big)$.

### C.1.3 When is CFM faster?

A practical rule from the above expressions:

$$\ell\Big(\tfrac{Rd}{m} + \tfrac{Rb}{m}\Big) + \ell b_{\max} \ < \ Rd + Rk.$$

For typical regimes with $d \gg b$ this simplifies to $\ell \frac{Rd}{m} < Rd + Rk$, i.e., $\frac{\ell}{m} < 1 + \frac{k}{d}$. Thus, once $m > \ell$ and $k$ is not tiny, CFM often has an advantage.

**Bottlenecks.** The path aggregation work in CFM is $\mathcal{O}(k+m)$ (each edge touched once), while its parallel time contribution is $\mathcal{O}(\ell b_{\max})$. Both are much smaller than projection/multiplication terms. The main limiter for speedups in practice is that many small matrix multiplies may utilize GPUs less efficiently than a few large ones.

**Example.** $R{=}512$, $d{=}768$, $k{=}100$, $m{=}50$, $b{\approx}3$, $\ell{\approx}\lceil\log_3 100\rceil{=}5$:

- CBM: $Rd + Rk = 393{,}216 + 51{,}200 = 444{,}416$

- CFM (parallel): $\ell\big(\frac{Rd}{m} + \frac{Rb}{m}\big) \approx 5\big(7{,}864 + 31\big) \approx 39{,}475$ (the extra $\ell b_{\max}$ is negligible here)

- Theoretical speedup $\approx 11.3\times$ (in practice smaller due to kernel efficiency).

**Early stopping (inference).** CFM can reduce effective work by pruning unlikely branches:

- Confidence thresholding: stop expanding once a path probability exceeds $\tau$

- Beam search: keep top-$B$ children per node

These heuristics reduce the effective depth/branching explored and thus the constant factors in the CFM terms.

