# OpenReview forum: "Concept Flow Models: Anchoring Concept-Based Reasoning with Hierarchical Bottlenecks"
_TMLR — Accepted by TMLR_

### Review · Reviewer_qzy6 · 2025-10-20

**Summary Of Contributions:**

The paper proposes Concept Flow Models (CFMs) which try to learn a meaningful feature space of human-understandable concepts, rendering the underyling decision processes interpretable. Unlike Concept Bottleneck Models (CBMs), which use a "flat" concept layer, CFMs assume a hierarchical concept structure that they model with a decision tree. The approach is supposed to increase interpretability by providing a sequence of increasingly fine-grained concepts as well as reduce concept leakage due sparsity of the tree structure.

**Pros:**
- well-written and good structure
- nice complexity analysis; nice theoretical analysis and new metric, given the paper's notion of "leakage"
- reasonable number of experimental settings and datasets


**Cons:**
- claims about the "automatic pipeline" downplay the required "manual review" used to ensure reasonable concept learning
- I'm still a bit confused about what "leakage" actually means here; it seems to be measured and limited via *capacity* rather than more directly focusing on meaningful concepts; would be nice with a more explicit definition of leakage, also discussing previous works like [1]
- empirical results seem to report only one run per setting rather than reporting standard errors or confidence intervals over a number of replicates

[1] Havasi, M., Parbhoo, S., & Doshi-Velez, F. (2022). Addressing leakage in concept bottleneck models. Advances in Neural Information Processing Systems, 35, 23386-23397.

**Audience:**

Yes

**Audience Explanation:**

Many independent lines of work boil down to learning structured concept spaces, and the paper here provides some new perspectives and techniques for the problem.

**Claims And Evidence:**

No

**Claims Explanation:**

The claims would be supported if the authors addressed the points mentioned in **Cons** above.

**Requested Changes:**

**Critical**: Address each of the **Cons** mentioned above, namely:
1. be more transparent in the abstract and intro about the limitations mentioned later in the paper, especially the "manual review" step, and the reliance on a sparse hierarchical underlying concept structure (which is a common and perhaps reasonable inductive bias, but still a strong and not-easily-testable assumption)
2. be a bit more explicit about what "leakage" means, for example with an (informal) definition and/or reference to previous literature; I do see that the authors attempted this, including referencing Srivastava et al. (2024) and the NEC metric, but the surrounding discussion (and e.g., Props 4.1 and 4.2) seems to me more related to complexity of the model class rather than other notions of concept leakage I've seen in the }literature; perhaps the authors can explain/clear this up for me, and nothing needs to be added to the paper?
3. the results should include some kind of measure of uncertainty, like standard errors or confidence intervals

---

> ### Author Response · Authors · 2025-11-28
> **Addressing Concerns on Transparency, Leakage Definition, and Uncertainty Measures**
>
> Dear Reviewer qzy6,
>
> We sincerely thank you for the thoughtful and constructive feedback. Below, we address each concern in detail.
>
> ---
>
> ## 1. Transparency in the Abstract and Introduction
>
> We acknowledge that the term "automatic pipeline" was not sufficiently precise and may have understated the role of manual review in our framework.
>
> In the interpretable machine learning community, manual intervention is a necessary and important interactive process to ensure alignment with human understanding. Specifically, CFM uses VLMs to extract hierarchical class structures, and the manual review step is essential to verify that the selected VLM backbone is appropriate for the given task. This is analogous to hyperparameter tuning or model selection in other machine learning pipelines.
>
> Additionally, we acknowledge that CFM operates under the assumption that classes exhibit hierarchical relationships. While this is a reasonable inductive bias for many real-world classification tasks (e.g., biological taxonomies, object hierarchies), it may not suit all scenarios.
>
> **Changes made:**
> - **Abstract:** Removed "automatic pipeline" claim and added scope limitation regarding hierarchical class structures.
> - **Introduction:** Revised contribution statements to more accurately reflect the methodology and assumptions.
>
> All changes are highlighted in **bold** in the revised manuscript and will be removed in the final version.
>
> ---
>
> ## 2. Definition of Information Leakage
>
> We appreciate the reference to Havasi et al. (2022). Our definition is consistent with established literature [1–4]: **information leakage occurs when the concept layer encodes information beyond the intended human-defined semantics**, allowing the model to achieve high accuracy through spurious correlations.
>
> To illustrate: imagine treating a disease requires drugs containing impurities (unintended information). Two mitigation strategies exist: (1) develop purer drugs, or (2) minimize the number of drugs needed while maintaining efficacy. Approach (1) corresponds to learning purer concepts; approach (2) corresponds selecting and using fewer concepts strategically.
>
> CFM, like several recent works [5–7], adopts the second approach. We accept that concept embeddings from CLIP inherently contain unintended information, and rather than purifying concepts directly, we use a hierarchical decision structure to limit concepts involved in each prediction. Our theoretical analysis (Propositions 4.1 and 4.2) characterizes this structural barrier:
> - **Proposition 4.1:** Random concepts require $m\times$ more concepts in CFM than CBM to achieve separability.
> - **Proposition 4.2:** Effective concepts per prediction scales as $O\left(\frac{R}{m} \log m\right)$ in CFM, substantially smaller than $O(R)$ for flat CBMs.
>
> **Changes made:**
> - **Introduction (first paragraph):** We have added an explicit definition of information leakage with references to prior work, including Havasi et al. (2022) and others [1–4].
> - **Introduction (second and third paragraphs):** We have clarified that CFM mitigates information leakage through the second approach, structural sparsity via hierarchical decision paths.
>
>
> ---
>
> ## 3. Measures of Uncertainty
>
> We fully agree that reporting uncertainty measures is essential for rigorous evaluation. Our original setup followed conventions of related works [5–7] using fixed random seeds.
>
> **we are currently repeating all empirical experiments supporting the main claims in our paper with multiple random seeds**. Once experiments are complete, we will update the manuscript and supplementary materials with mean values and standard deviations for all reported metrics.
>
> We hope the above explanations resolve the points you raised. We will notify you when the updated experimental results are available. Please let us know if there is anything further we should clarify or expand on.
>
> ---
>
> ## References
>
> [1] Havasi, M., et al. (2022). Addressing leakage in concept bottleneck models.
>
> [2] Mahinpei, A., et al. (2021). Promises and pitfalls of black-box concept learning models.
>
> [3] Margeloiu, A., et al. (2021). Do concept bottleneck models learn as intended?
>
> [4] Parisini, E., et al. (2025). Leakage and interpretability in concept-based models.
>
> [5] Yuksekgonul, M., et al. (2023). Post-hoc concept bottleneck models.
>
> [6] Yang, Y., et al. (2023). Language in a bottle.
>
> [7] Yan, A., et al. (2023). Learning concise and descriptive attributes for visual recognition.

---

> > ### Author Response · Authors · 2025-12-04
> > **Added Uncertainty Quantification (Updated Figure 3 & Table 1)**
> >
> > Dear Reviewer qzy6,
> >
> > We have repeated all experiments supporting Propositions 4.1 and 4.2 with three different random seeds, and updated Figure 3 and Table 1 accordingly. Specifically, Figure 3 now includes error bands, with detailed numerical values provided in Appendix B.4. These results robustly confirm CFM's structural resistance to random concept leakage. Table 1 now reports mean accuracy across runs; we omit standard deviations in the main table to preserve readability, but provide complete statistics in Appendix B.5. Our conclusions remain unchanged: CFM achieves comparable accuracy to CBM with significantly lower NEC (num of effective concepts) and higher SIR (Semantic Improvment over Random).
> >
> > Best regards,

---

> > > ### Comment · Reviewer_qzy6 · 2025-12-23
> > > **all concerns addressed**
> > >
> > > The author response and updated draft have addressed all of my original concerns. I now think that the paper supports all of its claims with accurate, convincing and clear evidence.

---

### Review · Reviewer_5oea · 2025-11-03

**Summary Of Contributions:**

TLDR: Well-known idea of hierarchy, applied to narrow domains on small size datasets with limited baselines; technically accurate but unclear how it advances the community.

Authors propose Concept Flow Models (CFMs), a hierarchical version of Concept Bottleneck Models (CBMs). Instead of exposing one big flat set of concepts to the classifier, they build a CLIP-based class hierarchy and make the model follow a root -> leaf path. At each node, only a small, node-specific set of concepts is used. The hierarchy is built automatically via agglomerative clustering of CLIP embeddings, pruned if nodes are not well separated, and then labeled with an LLM. For each node they select concepts using L1/Lasso from LLM-generated candidates. During inference, images are projected to these node-local concepts, node scores are turned into transition probabilities (temperature-scaled), and final class scores are aggregated along the path. Experiments on CIFAR-10/100, CUB-200, Tiny-ImageNet-200, and UCF101 show that this hierarchical bottleneck can match flat CBMs but with fewer effective concepts.

Strengths:

1. Paper targets an actual weakness of flat CBMs: when you expose all concepts at once, leakage becomes easy. Here they try to make leakage harder by only exposing concepts that are needed at each node. The CLIP-induced hierarchy -> node-local concepts -> path-based prediction pipeline is consistent with this goal.

2. There is some theoretical justification: decomposing the task into multiple node problems makes it more expensive (in terms of irrelevant/random info) to recreate the same leakage as in the flat setting. So the structure is not only heuristic, there is a rationale.

3. Authors compare their L1/Lasso-based node selector against other, more naive selection schemes (similarity-based, orthogonality-based, random, submodular) and show cleaner node-level concept sets. So the design choice is supported by experiments.

Weakness:

1. Evaluation is only on small/medium vision datasets. No ImageNet-1K, no modern long-tailed fine-grained sets, no domains where CLIP is weak (medical, satellite). So we don’t know if the hierarchical bottleneck still helps when classes explode or when text–vision alignment is worse.

2. Two very relevant hierarchical / leakage-related CBM baselines [1, 2] are missing. Both operate in almost the same problem space (CBM + hierarchy + leakage/intervention). Not comparing against them makes it harder to judge how much CFM actually moves the needle.

3. Strong dependence on CLIP and LLM quality. If the LLM proposes bad / non-visual concepts, or CLIP embeddings don’t match the label space, interpretability degrades. This will be worse in domains that are not web-image-like. Paper mentions this but doesn’t evaluate such a case.

Suggestion for improvement:

1. Show that the learned root -> leaf concept paths are useful beyond the classifier. For example, feed them as structured context to a VLM (similar to COCO-Tree style prompting) and check if hallucinations drop or VQA accuracy improves. That would make the hierarchy look like a general-purpose “video/image concept organizer,” not only an internal CBM trick.




References:

1. Sun, Ao, et al. "Eliminating information leakage in hard concept bottleneck models with supervised, hierarchical concept learning." arXiv preprint arXiv:2402.05945 (2024).
2. Pittino, Federico, Vesna Dimitrievska, and Rudolf Heer. "Hierarchical concept bottleneck models for vision and their application to explainable fine classification and tracking." Engineering Applications of Artificial Intelligence 118 (2023): 105674.

**Audience:**

Yes

**Audience Explanation:**

Yes. It’s relevant for TMLR readers working on interpretability and concept bottleneck models.

**Broader Impact Concerns:**

No broad impact concerns.

**Claims And Evidence:**

Yes

**Claims Explanation:**

Yes, but with gaps. For the presented scope, the empirical evidence supports the main claim: CFMs can match flat CBMs while reducing effective concept usage. For gaps, please check the weakness section.

**Requested Changes:**

Please address the weakness section.

---

> ### Author Response · Authors · 2025-11-28
> **Addressing Scalability, Related Work, and VLM Dependence**
>
> Dear Reviewer 5oea,
>
> We sincerely thank you for the thoughtful and constructive feedback. We address each concern below.
>
> ---
> ## 1. Scalability of CFM to Large-Scale Datasets
>
> We acknowledge that our experiments lack evaluation on large-scale datasets such as ImageNet-1K. As discussed in Section 3.4 and the Appendix, we've provided complexity analysis showing that CFM does not introduce significant computational overhead compared to CBMs during training and inference.
>
> The computational bottleneck arises from the concept selection phase (Section 3.3), specifically computing the similarity matrix $S_v \in \mathbb{R}^{n \times n_v}$ and solving Lasso regression over all $n$ training samples, which becomes prohibitive for datasets like ImageNet (Reviewer ZAYm also raised this concern in the Method section of their Requested Changes). CFM addresses scalability through two structural mechanisms: (1) the decision tree structure reduces the number of classes involved at each decision node, and (2) pruning prevents the tree from becoming excessively deep, keeping the number of internal nodes manageable.
>
> For the datasets with large num of samples, we do not need all samples to perform concept selection, a smaller subset suffices. We can apply **stratified sampling** at each node, bounding the number of samples (e.g., $n_{\text{max}} = 15,000$). This preserves the Lasso-based selection method while making computation tractable regardless of original dataset size.
>
> **Changes made:**
> - Section 3.3 adds this stratified sampling strategy for large-scale datasets.
> - We are conducting supplementary experiments on ImageNet-1K and will update the manuscript upon completion.
>
> All changes are highlighted in bold in the revised manuscript and will be removed in the final version.
>
> ---
> ## 2. Related Hierarchical CBMs
>
> We have added discussions of both papers in the revised Related Work section.
>
> **SUPCBM (Sun et al., 2024)** addresses *how to learn concept representations* that minimize leakage, introducing a two-level hierarchy *within concepts* and replacing the learnable predictor with a fixed intervention matrix. CFM addresses *how to organize existing concepts*: we construct a hierarchy *over classes* and distribute concepts to decision nodes. As discussed in our response to Reviewer qzy6 (Section 2), these correspond to complementary strategies—SUPCBM learns concept representations while CFM uses fewer concepts strategically.
>
> **Pittino et al. (2023)** applies CBMs to object detection with manually annotated concepts. Their "hierarchy" concatenates low-level concept categories into a flat vector—all concepts contribute equally. CFM's hierarchy operates over classes via a decision tree with path-specific concepts. Furthermore, Pittino et al. do not address information leakage, use no VLM backbone, and require manual annotation, making comparison impractical.
>
> ---
> ## 3. Dependence on CLIP and LLM Quality
>
> We acknowledge that CFM, like prior VLM-based CBMs [7-9], relies on vision-language model and LLM quality. We have strengthened this discussion in Sections 3.3 and 5.6. CFM's applicability will expand as domain-specific foundation models mature. Medical VLMs (MedCLIP [3], BiomedCLIP [4]) and remote sensing VLMs (RemoteCLIP [5], GeoRSCLIP [6]) demonstrate that vision-language alignment is increasingly available in specialized domains.
>
> ---
> ## 4. Generalization Beyond Classification
>
> Your suggestion to demonstrate CFM's utility beyond classification is very valuable. Additionally, CFM combines NBDT's tree structure with CLIP's zero-shot capabilities, which theoretically should benefit unseen classes and long-tailed fine-grained scenarios.
> While these are beyond the scope of our current contribution, we find this direction compelling and have noted it in the Conclusion section as promising future work.
>
> We hope these explanations resolve your concerns.  We will notify you when the updated experimental results are available.
>
> **References**
>
> [1] Sun, A. et al. (2024). Eliminating information leakage in hard concept bottleneck models with supervised, hierarchical concept learning.
>
> [2] Pittino, F. et al. (2023). Hierarchical concept bottleneck models for vision and their application to explainable fine classification and tracking.
>
> [3] Wang, Z. et al. (2022). MedCLIP: Contrastive learning from unpaired medical images and text.
>
> [4] Zhang, S. et al. (2023). BiomedCLIP: A multimodal biomedical foundation model.
>
> [5] Liu, F. et al. (2024). RemoteCLIP: A vision-language foundation model for remote sensing.
>
> [6] Zhang, Z. et al. (2024). RS5M and GeoRSCLIP: A large-scale vision-language dataset for remote sensing.
>
> [7] Yang, Y. et al. (2023). Language in a bottle: Language model guided concept bottleneck models for interpretable image classification.
>
> [8] Yan, A. et al. (2023). Learning concise and descriptive attributes for visual recognition.
>
> [9] Shang, C. et al. (2024). Incremental residual concept bottleneck models.

---

> ### Author Response · Authors · 2025-12-06
> **Addressing Scalability of CFM to Large-Scale Datasets**
>
> Dear Reviewer 5oea,
>
> We have now finished our experiments on ImageNet-1K.
>
> The computational bottleneck arises from computing the similarity matrix $S_v \in \mathbb{R}^{n \times n_v}$ and solving Lasso regression over all $n$ samples in concept selection (Section 3.3). We address this by applying a simple trick without modifying the CFM pipeline, namely using stratified sampling: grouping samples by class and drawing a balanced subset.(e.g., $n_{\max} = 15{,}000$), as we don't need all samples to perform concept selection.
>
> Results are summarized below (mean ± std over 3 runs):
>
> | Method | NEC | Acc (%) | SIR (%) |
> |--------|-----|---------|---------|
> | Linear Probe | N/A | 73.03 ± 0.63 | N/A |
> | CBM (Random) | 853.92 ± 16.70 | 72.82 ± 0.43 | 0 |
> | CFM (Random) | 260.10 ± 1.62 | 69.61 ± 0.23 | 0 |
> | PCBM | 860.44 ± 8.18 | 73.08 ± 0.76 | 0.37 ± 1.51 |
> | Labo | 868.31 ± 0.69 | 72.86 ± 0.55 | 0.06 ± 0.42 |
> | CFM (Ours) | **339.18 ± 1.12** | **72.88 ± 0.56** | **4.70 ± 0.58** |
>
> CFM scales effectively to ImageNet-1K (1000 classes, 1.2M images), achieving competitive accuracy (72.88%) with significantly fewer effective concepts (339 vs. ~860). With 1000 concepts exceeding the 512-dimensional embedding, flat CBMs exhibit severe information leakage. CBM (Random), PCBM, and Labo all fall within 0.25% of Linear Probe, with near-zero SIR confirming that random concepts suffice when the concept count exceeds the embedding dimension. In contrast, CFM with 5 internal nodes and 3 levels limits per-node concepts below the embedding dimension, yielding higher SIR (4.70%) and validating hierarchical bottlenecks' resistance to information leakage at scale.
>
> **Changes made:** Section 3.3 describes stratified sampling; Appendix B.8 now includes ImageNet-1K results.
>
> Best regards,

---

> > ### Comment · Reviewer_5oea · 2025-12-07
> > **Response**
> >
> > Overall, I feel that the authors have satisfactorily addressed my main concerns. The new ImageNet-1K experiments resolve the scalability question, and the expanded Related Work clarifies how CFM relates to prior hierarchical/leakage-aware CBMs. The discussion of CLIP/LLM dependence is now more explicit, with remaining limitations clearly framed as future work. Given these updates, I am broadly satisfied with the current revision.

---

### Review · Reviewer_ZAYm · 2025-11-24

**Summary Of Contributions:**

## **Summary**
This paper explores how to mitigate concept leakage in concept-based interpretable models when the concept set is large (particularly when it is larger than the embedding dimensionality). It achieves this by introducing the Concept Flow Model (CFM), a hierarchical concept-based architecture that splits the predictive process into a hierarchy of prediction sub-problems, each mapping concept subsets to intermediate semantic nodes and generating paths that lead to one or more leaf nodes representing output task classes. CFM operates in three steps. First, it uses hierarchical clustering of class-level embeddings to produce a tree whose leaf nodes are task classes and whose intermediate nodes are given human-like interpretations with the aid of a Foundation model. Second, it uses a label-free pipeline to produce concept candidates and splits those concepts across intermediate nodes in the hierarchy by solving a Lasso selection problem at each node to select a sparse set of informative subconcepts. Finally, it learns transition probabilities between intermediate nodes, allowing one to assign probabilities to each path in the tree and to learn the model’s parameters via MLE. This paper shows, both **theoretically** and **empirically**, that CFMs avoid leaking information of concepts outside their selected predictive path and, by doing so, lead to high-performing models that, compared to existing label-free baselines, (1) have a better effective concept size, (2) can achieve better or competitive task accuracies, and (3) can generate human-interpretable hierarchical explanations for their downstream predictions.

## **Strengths And Weaknesses**

Thank you so much for submitting this work! I enjoyed reading this paper. Below are what I believe are this paper’s main strengths, followed by what I think are some of its weaknesses:

### Strengths

1. **[Significance, Critical]** Overall, I think the paper’s framing of how to address information leakage via hierarchical modeling, and the learning algorithm that follows, is very interesting and theoretically sound. More importantly, I think the proposed mechanism for generating, naming, and learning transition probabilities in CFM’s intermediate nodes can be easily extended and applied in other fields and directions beyond information leakage. Therefore, I believe this work has the potential to attract not just the concept-based interpretability community but also the neurosymbolic and reasoning communities.
2. **[Quality, Critical]** The presence of both non-trivial theoretical and empirical results makes this paper’s evaluation very strong. In particular, I appreciate how the experiments were carefully divided and explained, how individual components of the architecture were tested via ablations, and how the paper presents qualitative results at the end. Moreover, I want to mention that I appreciate this paper including a limitations section and an extensive set of appendices that facilitate the reproducibility of the results.
3. **[Clarity, Major]** The writing is very clear and easy to read. I have some issues with the specific sections that lack breaks in long paragraphs (as described below). Still, aside from that, the paper was very easy to follow (even the theoretical parts, which were written extremely well).
4. **[Originality, Minor]** The splitting of prediction into n-ary trees that yield different predictive paths using different concept subsets is certainly novel and an interesting way of approaching the issues that arise from leakage when one deals with large concept sets. The only reason I do not mark this as a critical strength is that several related ideas have been explored in the past (and should, at the very least, be discussed in the paper, as I discuss below).

### Weaknesses

In contrast, I believe the following are some of this work’s limitations:

1. **[Quality, Critical]** As discussed below in my requested changes, I have some trouble with the main results table (Table 1). In particular, I am not entirely convinced that it is fair to compare against existing methods without using those methods as defined by the authors (e.g., using Labo or PCBM without sparsity regularizers). Please see below for specific questions on this matter.
2. **[Quality, Critical]** Against good conventional scientific practices, there are no error bars included for any of the empirical results. This makes it difficult to judge the significance of any observed differences.
3. **[Quality, Major]** The qualitative study in the paper lacks depth (there is only a single example across the entire paper, including the appendix).
4. **[Clarity, Minor]** Some of the writing could be improved by splitting long paragraphs into more processable bits of text. See below for specific feedback on this issue.
5. **[Originality, Minor]** The idea of exploring non-flat predictive concept-to-label models is not entirely novel. In fact, this has been explored by several previous works (e.g., [1-7] to name only a few). Because of this, this direction is not entirely novel. That being said, I mark this weakness as minor for two reasons: (1) novelty, as long as findings are sound and interesting, is not one of TMLR’s key criteria; and (2) the appoach used is relatively different to those used in the past, although similar ideas have been explored also with leakage as a motivating factor [7].

### References

1. Havasi et al. "Addressing leakage in concept bottleneck models." NeurIPS (2022)
2. Xu et al. "Energy-based concept bottleneck models: Unifying prediction, concept intervention, and probabilistic interpretations." ICLR (2024).
3. Raman et al. "Understanding inter-concept relationships in concept-based models." ICML (2024).
4. Vandenhirtz et al. "Stochastic concept bottleneck models." NeurIPS (2024)
5. Dominici et al. "Causal concept graph models: Beyond causal opacity in deep learning." ICLR (2025).
6. Barbiero et al. "Relational concept bottleneck models." NeurIPS (2024).
7. Ragkousis et al. "Tree-based leakage inspection and control in concept bottleneck models." *arXiv:2410.06352* (2024).

**Audience:**

Yes

**Audience Explanation:**

As described in my summary, I believe this paper will be of interest for the XAI/Interpretability communities (due to its use of a concept-unsupervised pipeline for learning hierarchical explanations) and the neurosymbolic/reasoning communities (due to the generality of the proposed algorithm for learning a human-understandable hierarchy of intermediate reasoning steps for a downstream task of interest). Therefore, I believe a large community within TMLR's audience will find this paper interesting.

**Broader Impact Concerns:**

I do not believe there are any broader ethical implications from this work that would merit a Broader Impact Statement.

**Claims And Evidence:**

Yes

**Claims Explanation:**

The paper discusses both theoretical and empirical results. The theoretical results appear sound and are empirically verified in the first section of the experimentation. The rest of the empirical results then provide clear evidence that the proposed pipeline can learn to accurately model a downstream task of interest while utilizing a relatively small effective concept size and producing helpful hierarchical explanations for its predictions. Moreover, the experimental section includes ablations that provide clear evidence that each component in the proposed method serves its intended purpose within the architecture. See my summary above for further details.

**Requested Changes:**

Below, I list some potential changes for each section and their importance in securing my recommendation for acceptance. If time is limited, please focus on all **critical** requests, followed by all **major** requests. If I misunderstood something at any point, just let me know, as this is always possible (apologies in advance if that’s the case!).

### Section 1 (Introduction)

- **[Major, Attribution]** Information leakage was first discussed by Mahinpei et al. [1] and by Margeloiu et al. [2]. At the very least, I would strongly suggest citing these works when information leakage is first introduced.
- **[Minor, Clarity]** The first paragraph of the introduction is very large (it spans more than half a page). I would suggest breaking it up into more manageable chunks.

### Section 2 (Related Work)

- **[Major, Missing References]** The following are missing references that I would consider including in this related work, as they are highly relevant to subsections of your related work and are not discussed anywhere:
    - Espinosa Zarlenga et al. "Concept embedding models: Beyond the accuracy-explainability trade-off." NeurIPS (2022)
    - Havasi et al. "Addressing leakage in concept bottleneck models." NeurIPS (2022).
    - Espinosa Zarlenga et al. "Learning to receive help: Intervention-aware concept embedding models." NeurIPS (2023)
    - Rodriguez et al. "On the fusion of soft-decision-trees and concept-based models." Applied Soft Computing (2024)
    - Ruiz Luyten et al. "A theoretical design of concept sets: improving the predictability of concept bottleneck models." NeurIPS (2024)
    - Ragkousis et al. "Tree-based leakage inspection and control in concept bottleneck models." arXiv (2024).
    - Vandenhirtz et al. "Stochastic concept bottleneck models." NeurIPS (2024)
    - Parisini et al. "Leakage and interpretability in concept-based models." arXiv (2025).
    - Dominici et al. "Causal concept graph models: Beyond causal opacity in deep learning." ICLR (2025).
- **[Minor, Clarity]** As in Section 1, this entire section is a single paragraph. For clarity and readability, I suggest breaking it into multiple paragraphs.

### Section 3 (Method)

- **[Critical, Clarity]** I would strongly suggest not using $c$ to refer to task classes when you have also “concepts” all over the place in the methodology itself. This opens up the door for extremely confusing notation. Instead, I would recommend using something like $l$ (for label) to refer to task labels (rather than for a path as it is currently used) and $c$ to refer to intermediate concepts. Also, on this note, notice that $k$ is usually used for the cardinality of the concept space (rather than the cardinality of the task space), so using $K$ for the number of task classes also invites confusion for those of us used to different notation in the concept-based literature.
- **[Major, Quality]** In Section 3.2, it would be helpful if, in this section, rather than in an appendix, you clarify how sensitive the method is to different values of $\{t, \tau, r_\text{min}, \alpha\}$ (and whether that selection matters). Ideally, this is backed with some further experiments down the line (in the experimentation section or an appendix). I can see a discussion of some of these hypers in the appendices, but I could not find any discussion of the method's sensitivity to these values.
- **[Major, Clarity]** Given that this is a relatively important component of your pipeline, and one that may become a serious practical constraint on the usability of your proposed method, I would recommend discussing the “manual review”  mentioned at the end of Section 3.2 within the main body of the paper rather than in an appendix.
- **[Major, Quality]** How well does this method scale for large datasets, given that computing Equation 2 could be quite computationally heavy (if not intractable) for very large datasets? I would recommend that the authors discuss ways to mitigate this potential issue in Section 3.3, when the concept generation/filtering mechanism is defined.
- **[Major, Clarity]** The variable $B$ is used in Equation 4 but not defined anywhere up to this point in the manuscript. Please define this variable before or immediately after using it.
- **[Minor, Notation]** Note that there are inconsistencies in how this paper names the children of a node (both $\text{Ch}(v)$ and $\text{Child}(v)$ are used). I would suggest sticking with one convention only.

### Section 5 (Experiments)

- **[Critical, Quality]** I can see an argument for why, in some experiments, you may want to evaluate against PCBMs and Labos without sparsity regularization. However, when comparing it against your method on the purely-accuracy-based experiments (i.e., in Section 5.3), I would argue that you should compare with, at least, the original variant of both of these approaches as proposed by the authors (which includes the sparsity regularization). I can see that some of these experiments are in Appendix B.3. However, I would argue that it is important to show the results of using sparsity regularization for Labo and PCBM in Table 1 as that is how the authors proposed those methods (otherwise calling them Labo/PCBM in the table itself is a bit misleading and may give the wrong impression). After doing this, and as a follow-up experiment, and this is where fairness comes in place, you can then evaluate what happens when one includes or excludes regularization to all methods (to see if really going over the decision-tree path is worth it over simpler methods such as sparsity regularization). This is important as I am not entirely convinced by the argument in Appendix B.3 that fine-tuning for regularization is somehow harder than having to fine-tune for the hyperparameters that are now present in CFM but are not in other methods (e.g., $t$, $\tau$, $\alpha$, $r_\text{min}$, etc.). For example, why is it harder to select $\lambda$ in the sparsity regularization than it is to select $\alpha$ for CFM? I am just not entirely convinced that they are inherently different problems. Therefore, I would greatly appreciate it if the authors could further comment on this.
- **[Critical, Quality]** I would greatly appreciate it if error bars could be included for all empirical results to follow good scientific practice. This is important because some results are close enough to be due to noise.
- **[Major, Quality]** Regarding the many claims in the paper of CFM reducing information leakage: would it be possible to measure this (to see if the claim holds) against other more information-theoretic metrics for leakage (e.g., the ones in [3] or [4]) than the proposed SIR metric?
- **[Minor, Clarity]** I would suggest splitting the first paragraph of Section 5.4 into two. One part can discuss the results of Table 2, and the other can discuss Figure 4. Right now, these two experiments are discussed in the same paragraph, with little to no connection between them. A similar comment applies to the paragraph in Section 5.5.
- **[Nit/Minor, Typo]** There is a space missing in "PCBM(Yuksekgonul et al., 2022)” and in “Labo(Yang et al., 2023)” in Section 5.3.

### General Questions

- **[Critical, Quality]**  Do you have more qualitative examples showing hierarchical explanations for downstream tasks? Having some extra examples could help clarify how helpful this method's explanations are in practical situations (rather than showing a simple example where it might've gotten "lucky" with its explanation).
- **[Major, Quality]** Could you please provide a comparison somewhere in the text (at least conceptual, so not necessarily empirical) of CFMs vis-a-vis methods that also consider cross-concept relationships when making their predictions, such as Autoregressive CBMs [5], Stochastic CBMs [6], or Concept Graph Models [7]?

## References

1. Mahinpei et al. "Promises and pitfalls of black-box concept learning models." arXiv (2021).
2. Margeloiu et al. "Do concept bottleneck models learn as intended?" arXiv (2021).
3. Zarlenga Espinosa, et al. "Towards robust metrics for concept representation evaluation." AAAI (2023).
4. Parisini et al. "Leakage and interpretability in concept-based models." arXiv (2025).
5. Havasi et al. "Addressing leakage in concept bottleneck models." NeurIPS (2022)
6. Vandenhirtz et al. "Stochastic concept bottleneck models." NeurIPS (2024)
7. Dominici et al. "Causal concept graph models: Beyond causal opacity in deep learning." ICLR (2025).

---

> ### Author Response · Authors · 2025-12-02
> **Addressing Concerns related to Sections Introduction, Related Work and Method**
>
> Dear Reviewer ZAYm,
>
> Thank you for your positive evaluation of our work and constructive suggestions. Due to the character limit, we first address concerns related to Introduction, Related Work, and Method sections. Remaining concerns will be addressed in the following response.
>
> All changes are highlighted in bold in the revised manuscript and will be removed in the final version.
>
> ## Section 1: Introduction
>
> **[Major, Attribution]** We have added citations to Mahinpei et al. and Margeloiu et al. when introducing "information leakage."
>
> **[Minor, Clarity]** We have split the introduction into four paragraphs for better readability and smoother transitions.
>
> ## Section 2: Related Work
>
> **[Major, Missing References]** Thank you for providing these highly relevant works. We have split this section into three paragraphs and added more related work and comparison: (1) works on enhancing concept representations, concept intervention, and the interpretability-accuracy tradeoff; (2) existing approaches addressing information leakage; (3) works on decision trees and hierarchical concepts, including discussion of their connections and comparisons with our approach.
>
> **[Minor, Clarity]** Addressed by the paragraph restructuring above.
>
> ## Section 3: Method
>
> **[Critical, Clarity] Notation:** We agree to revise the notation for task class indices and the number of classes to follow conventions in the concept-based literature.
>
> 1. **Class indices:** We now use $l$ (for "label" or "leaf") instead of $c$: e.g., $y_{il} = \mathbb{I}\lbrack y_i = l \rbrack$ and $\sum_{l=1}^{|L|}$.
> 2. **Number of classes:** We replace $K$ with $|L|$, leveraging the fact that leaf nodes $L \subset V$ directly correspond to target classes.
> 3. **Class centroids:** We rename $c_i$ to $\mu_i$, with the set $\mathcal{M} = \lbrace \mu_1, \dots, \mu_{|L|} \rbrace$.
> 4. **Concept notation:** $C_v$ continues to denote concept matrices, now unambiguously distinguished from class-related variables.
>
> **[Major, Clarity] Variable $B$:** We now define $B$ (average concept share per node) before using it in Equation 4.
>
> **[Minor, Notation] Ch(v) vs Child(v):** We now use $\mathrm{Ch}(v)$ consistently throughout the paper.
>
> **[Major, Clarity] Manual review:** As noted in our response to Reviewer rqzy, we believe manual review is a necessary and important interactive process in interpretable machine learning to ensure alignment with human understanding. CFM uses VLMs to extract hierarchical class structures, and the manual review step verifies that the selected VLM backbone is appropriate for the given task. We have expanded the description of manual review in Section 3.2 to emphasize its role.
>
> **[Major, Quality] Hyperparameters $t$, $\tau$, $r_{\min}$, $\alpha$:** We first clarify that $t$ and $\tau$ refer to the same hyperparameter: $\tau$ is the $t$-th percentile of merge distances. We have revised the text to use only $t$ to avoid confusion.
>
> These hyperparameters are inherently coupled with the **manual review step**, functioning more as architecture design choices than traditional hyperparameters:
> - **$t$:** Controls the number of internal nodes. Figure 4 demonstrates the effect of tree depth on accuracy.
> - **$\alpha$ and $r_{\min}$:** Control concept allocation per node. For unbalanced trees, higher $\alpha$ allocates more concepts to high-branching nodes.
>
> We have added clarifications in Section 3.3 regarding parameter selection.
>
> **[Major, Quality] Scalability:** Thank you for raising this concern, which was also noted by Reviewer 5oea. We address the computational overhead of Equation 2 via **stratified sampling**: since our goal is to select representative concepts rather than train a model, using a smaller subset (e.g., $n_{\max} = 15,000$) is sufficient for effective Lasso-based selection while remaining tractable regardless of dataset size. We have added this description in Section 3.3. We are currently conducting experiments on ImageNet-1K to further validate scalability; we will update the manuscript once results are available.

---

> > ### Author Response · Authors · 2025-12-02
> > **Addressing concerns related to the Experiments and General Questions**
> >
> > Dear Reviewer ZAYm,
> >
> > In this response, we continue to address your concerns.
> >
> > ---
> >
> > ## Section 5: Experiments
> >
> > **[Critical, Quality]**
> >
> > We fully appreciate your concern regarding the comparison with Labo/PCBM in Table 1.
> >
> > **Clarification on Labo:** The original Labo implementation does not apply sparsity regularization. We re-examined their paper and the provided code repository—the paper does not mention sparsity regularization, and in their code this option is set to `False`. Therefore, the comparison between CFM and Labo is fair.
> >
> > **Clarification on PCBM:** In our supplementary material, we did implement elastic net regularization for PCBM but set it to `False`. The purpose of Table 1 is to empirically support Proposition 4.2—that CFM can achieve comparable accuracy to flat CBMs while using fewer effective concepts. Introducing sparsity regularization would confound attribution: we could not distinguish whether improvements stem from CFM's structural design or from regularization. Moreover, as noted in Appendix B.3 and VLG-CBM (Srivastava et al., 2024), fair comparison requires tuning λ to match NEC across methods, which is why we disabled regularization for PCBM in Table 1.
> >
> > **On hyperparameter complexity:** The tuning complexity is indeed comparable: CFM's structural sparsity is controlled by the number of internal nodes ($t$) and the concept allocation ratio ($\alpha$), while sparsity regularization requires tuning the regularization strength ($\lambda$) and the L1/L2 ratio. Importantly, structural sparsity and sparsity regularization are not mutually exclusive—they can be combined, as we demonstrate below.
> >
> > **Additional experiments:** Following your suggestion, we conducted additional experiments on CIFAR-10 comparing PCBM and CFM with and without sparsity regularization. For experiments with regularization, we set the L1 ratio to 0.99 (matching the original PCBM setting) and tuned $\lambda$ to achieve matched NEC values of 5 and 3, following the methodology in VLG-CBM:
> >
> > | Method | Sparsity Reg. | NEC | Accuracy (%) | $\lambda$ |
> > |--------|---------------|-----|--------------|-----------|
> > | PCBM   | ✗             | 57  | 91.36        | —         |
> > | CFM    | ✗             | 7   | 91.68        | —         |
> > | PCBM   | ✓             | 5   | 85.85        | 0.8       |
> > | CFM    | ✓             | 5   | 90.66        | 0.02      |
> > | PCBM   | ✓             | 3   | 84.41        | 1.3       |
> > | CFM    | ✓             | 3   | 90.61        | 0.1       |
> >
> > These results demonstrate that: (1) without sparsity regularization, CFM achieves comparable accuracy with substantially fewer effective concepts; (2) with sparsity regularization at matched NEC levels, CFM significantly outperforms PCBM; and (3) CFM's structural sparsity and sparsity regularization can be effectively combined. We will update Appendix B.3 with these results.
> >
> > ---
> >
> > **[Critical, Quality]**
> >
> > As noted in our response to Reviewer qzy6, we are currently repeating all empirical experiments supporting the main claims in our paper with multiple random seeds. Once experiments are complete, we will notice you.
> >
> > ---
> >
> > **[Major, Quality]**
> >
> > As noted in our response to Reviewer qzy6 about the definition of information leakage, which occurs when the concept encodes information beyond the intended human-defined semantics, allowing the model to achieve high accuracy through spurious correlations. There are two lines of approaches to mitigate information leakage: (1) learning purer concepts; (2) selecting and using fewer concepts strategically. Our work aligns with the second line of approaches, and we indeed included the metric **NEC** (number of effective concepts) proposed by Srivastava et al. (2024) in addition to our metric **SIR**. The information-theoretic metrics you mentioned in previous works are designed for the first line of approaches, which measure the purity of concept representations—this is orthogonal to our work.
> >
> > ---
> >
> > **[Minor, Clarity]**
> >
> > We have split Section 5.4 and 5.5 for better readability and transition.
> >
> > ---
> >
> > **[Nit/Minor, Typo]**
> >
> > We have corrected the missing space in citations in Section 5.3 (e.g., "PCBM (Yuksekgonul et al., 2022)" and "Labo (Yang et al., 2023)").
> >
> > ---
> >
> > ## General Questions
> >
> > **[Critical, Quality]**
> >
> > We agree that we should provide more examples for qualitative results. The initial version provided only two examples from CIFAR-10 and CUB-200 due to the page limit required by submission. We are now including more examples from other datasets in the Appendix and will notice you once we have updated the manuscript.
> >
> > ---
> > **[Major, Quality]**
> > We have revised the first paragraph of the Related Work section to include a conceptual comparison with Autoregressive CBMs (Havasi et al., 2022), Stochastic CBMs (Vandenhirtz et al., 2024), and Causal Concept Graph Models (Dominici et al., 2025). The manuscript has been updated accordingly.
> >
> >
> > ---
> > We hope these clarifications and additional experiments address your concerns.

---

> > > ### Author Response · Authors · 2025-12-06
> > > **Additional Experiments on Imagenet-1k and Qualitative Analysis**
> > >
> > > Dear Reviewer ZAYm,
> > >
> > >
> > > We have now finished experiments on ImageNet-1K, comparison with/without sparsity regularization, and qualitative analysis on additional examples. We address each concern as follows:
> > >
> > > **1) Qualitative Analysis of Decision Paths (Appendix B.7)**
> > >
> > > We have added four detailed examples across datasets (three correct classifications and one failure case) to demonstrate CFM's explanatory utility.
> > > For correct predictions, the stepwise decision paths show how concepts progressively narrow the prediction scope (e.g., Root → Sports and Athletic Events → bowling), with each transition utilizing semantically relevant concepts. The failure case (parking meter misclassified as reel) demonstrates CFM's diagnostic capability: the uncertain routing at the second transition (69.24% vs. 30.76%) and semantically misaligned concepts provide interpretable signals of unreliable predictions, something flat CBMs cannot offer through their single linear combination.
> > >
> > > We also acknowledge that even correctly classified samples may activate semantically incorrect concepts (e.g., "colored cue ball markings" for bowling), reflecting CLIP-based alignment limitations noted in prior work. This reinforces that concept pool quality remains important for explanation fidelity regardless of architecture.
> > >
> > > **2) Scalability to Large-Scale Datasets (Appendix B.8)**
> > >
> > > The computational bottleneck arises from computing the similarity matrix $S_v \in \mathbb{R}^{n \times n_v}$ and solving Lasso regression over all $n$ samples in concept selection (Section 3.3). We address this by applying a simple trick without modifying the CFM pipeline, namely using stratified sampling: grouping samples by class and drawing a balanced subset.(e.g., $n_{\max} = 15{,}000$), as we don't need all samples to perform concept selection.
> > > ImageNet-1K experiments validating the scalability are now included in Appendix B.8.
> > >
> > > **3) Comparison with Sparsity Regularization (Appendix B.6)**
> > >
> > > Comparison of CFM and PCBM with/without sparsity regularization has been added in Appendix B.6.
> > >
> > > **Changes made:** Appendices B.6, B.7, and B.8 added; Section 5 updated accordingly.
> > >
> > > Best regards,

---

> > > > ### Comment · Reviewer_ZAYm · 2025-12-24
> > > >
> > > > Dear Authors,
> > > >
> > > > Thank you so much for your very careful rebuttal and answers to all my review's concerns. I can confirm that the new manuscript and experiments, together with your clarifications provided above, appropriately address all my concerns. I believe the paper is much stronger now, and I wish you the best of luck with this submission.

---

### Decision · Action_Editor_s4yk · 2025-12-24

**Recommendation:** Accept as is

**Audience:**

Yes

**Audience Explanation:**

Concept bottleneck models are clearly of general interest, and the proposal in this paper addresses a known issue with CBMs.

**Claims And Evidence:**

Yes

**Claims Explanation:**

While some concerns were raised initially, after revision these concerns were satisfactorily addressed. Novelty is a lingering concern from some reviewers, however, novelty is not a consideration for TMLR.